# Connectivity-Driven Pseudo-Labeling Makes Stronger Cross-Domain Segmenters

**Dong Zhao**[1*]**, Qi Zang**[1*]**, Shuang Wang**[1†]**, Nicu Sebe**[2]**, Zhun Zhong**[3,4†]

[1] School of Artificial Intelligence, Xidian University, Shaanxi, China
[2] Department of Information Engineering and Computer Science, University of Trento, Italy
[3] School of Computer Science and Information Engineering, Hefei University of Technology, China
[4] School of Computer Science, University of Nottingham, NG8 1BB Nottingham, UK

## Abstract

Presently, pseudo-labeling stands as a prevailing approach in cross-domain semantic segmentation, enhancing model efficacy by training with pixels assigned with reliable pseudo-labels. However, we identify two key limitations within this paradigm: (1) under relatively severe domain shifts, most selected reliable pixels appear speckled and remain noisy. (2) when dealing with wild data, some pixels belonging to the open-set class may exhibit high confidence and also appear speckled. These two points make it difficult for the pixel-level selection mechanism to identify and correct these speckled close- and open-set noises. As a result, error accumulation is continuously introduced into subsequent self-training, leading to inefficiencies in pseudo-labeling. To address these limitations, we propose a novel method called Semantic Connectivity-driven Pseudo-labeling (SeCo). SeCo formulates pseudo-labels at the connectivity level, which makes it easier to locate and correct closed and open set noise. Specifically, SeCo comprises two key components: Pixel Semantic Aggregation (PSA) and Semantic Connectivity Correction (SCC). Initially, PSA categorizes semantics into "stuff" and "things" categories and aggregates speckled pseudo-labels into semantic connectivity through efficient interaction with the Segment Anything Model (SAM). This enables us not only to obtain accurate boundaries but also simplifies noise localization. Subsequently, SCC introduces a simple connectivity classification task, which enables us to locate and correct connectivity noise with the guidance of loss distribution. Extensive experiments demonstrate that SeCo can be flexibly applied to various cross-domain semantic segmentation tasks, *i.e.* domain generalization and domain adaptation, even including source-free, and black-box domain adaptation, significantly improving the performance of existing state-of-the-art methods. The code is available at https://github.com/DZhaoXd/SeCo.

## 1 Introduction

Propelled by deep neural networks, remarkable strides have been achieved in semantic segmentation technology [8, 10, 80, 31]. However, deep segmentation models encounter a significant decline in adaptability when confronted with open domains. This challenge is mainly attributed to the inherent domain shift between the training and testing data [22, 9, 13]. To address cross-domain challenges, Domain Adaptation (DA) [21] and Domain Generalization (DG) [11] techniques have been proposed to enhance the segmenter's adaptability to the target domain or unseen domains.

Pseudo-labeling [67] is a widely used technique in cross-domain semantic segmentation tasks, enhancing model efficiency by training with pixels assigned reliable pseudo-labels. The core of

---

[*]Equal contribution.     [†] Corresponding authors.

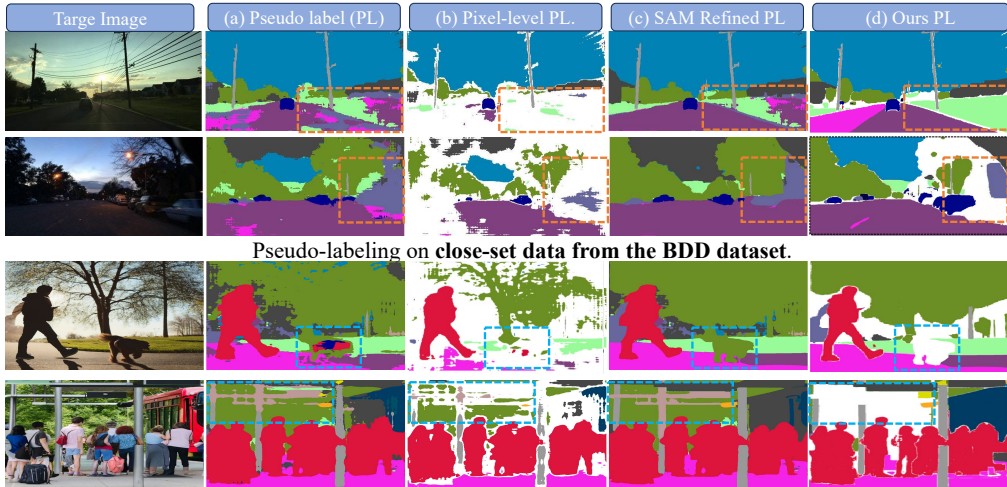

| Target Image | (a) Pseudo label (PL) | (b) Pixel-level PL. | (c) SAM Refined PL | (d) Ours PL |

Pseudo-labeling on **close-set data from the BDD dataset**.

Pseudo-labeling on **open-set data synthesized from stable diffusion**.

Figure 1: Comparison of (a) Pseudo-Labels (PL), (b) pixel-level PL [39], (c) SAM-refined PL [4], and (d) the proposed connectivity-level PL. The white area in the PL represents the filtered area. Our method effectively filters out and corrects closed-set noise (the orange box) induced by domain shifts, as well as open-set noise (the cyan box) in the wild data (*e.g.*, synthesized from stable diffusion [64]).

pseudo-labeling is how to eliminate semantic noise. In the DA task, various works are dedicated to designing efficient selection or training methods, such as multi-classifier voting [97, 90] or augmentation consistency [2, 26] principles to stabilize noisy training. Furthermore, in the DG task, advanced work [4] has also shown that pseudo-labeling can be used to leverage in-the-wild data synthesized by stable diffusion, enhancing the segmenter's generalization to unseen domains.

Despite the significant advancements made by these methods, we have identified limitations in the pseudo-labels as depicted in Fig. 1. *Firstly*, under relatively severe domain shifts, most selected reliable pixels appear speckled and remain noisy. (the orange box). *Secondly*, when dealing with in-the-wild data, pixels belonging to the open-set class may also be selected as 'reliable' and still exhibit speckle (the cyan box). These make it difficult for the selection mechanism to identify and correct these speckled close- and open-set noises. As a result, speckle noise labels with open-set or closed-set noise are introduced into the subsequent self-training process, leading to severe error accumulation. These issues indicate that constructing pixel-level uncertainty measures to filter noisy pseudo-labels is very challenging, especially in open environments.

Segment Anything Model (SAM) [37] is a foundation segmentation model that takes both images and geometric prompts (points, boxes, masks) as input and outputs class-agnostic masks. Motivated by this, some attempts [4, 5] have been made to adopt SAM to refine the pseudo-labels. For instance, using reliable pixel pseudo-labels to prompt SAM to generate class-agnostic masks, and then assigning pseudo-classes to these masks. However, due to inappropriate prompting and semantic noise, these attempts may not improve the quality of pseudo-labels and even exacerbate their semantic noise, as shown in Fig. 1(c).

In this paper, we introduce a novel method called Semantic Connectivity-driven Pseudo-labeling (SeCo) for cross-domain semantic segmentation. SeCo models the distribution of pseudo-label noise at the connectivity level, allowing for effective correction of closed-set noise and removal of open-set noise. SeCo comprises two key components: Pixel Semantic Aggregation (PSA) and Semantic Connectivity Correction (SCC). Initially, PSA splits the categories into the "stuff" and "things" forms. Then, PSA efficiently aggregates speckled pseudo-labels into semantic connectivity [2] by interacting with the Segment Anything Model (SAM). This strategy not only ensures precise boundaries but also streamlines noise localization, as distinguishing noise at the connectivity level is inherently more straightforward than at the pixel level. Subsequently, SCC introduces a sample connectivity classification task for learning connectivity with noisy labels. As connectivity classification focuses on local overall categories, we propose to leverage the technique of *early learning* in noisy label

---

[2]We refer to the concept of "connected components" in traditional image processing [30], and call the connected regions here as "semantic connectivity".

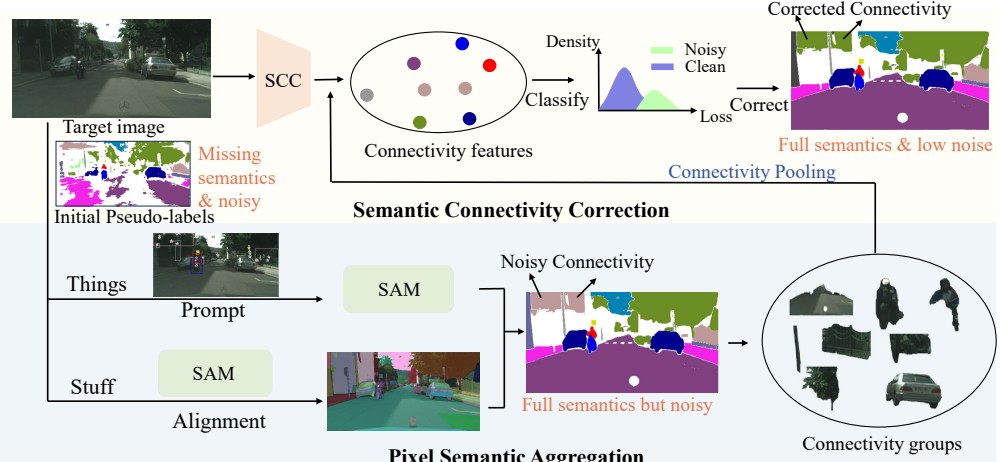

Figure 2: The pipeline of the proposed Semantic Connectivity-Driven Pseudo-labeling (SeCo). In (a), pixel-level pseudo-labels are interactively aggregated into connectivity by SAM using the "stuff and things" manner, grouping semantically similar pixels. Then, in (b), these connectivities are treated as classification objects and are identified for semantic noise by noisy learning. This process is handled offline, and the corrected high-quality pseudo-labels can be used for further self-training.

learning [88, 45] to identify connectivity noise, guided by loss distribution. As illustrated in Fig. 1(d), the incorporation of the proposed connectivity-driven pseudo-labels significantly enhances the quality of pseudo-labels, showcasing complete structures and reduced close- and open-set category noise.

In summary, the contributions of this paper are threefold. **First**, we identify the drawbacks of the pseudo-labeling technique and highlight the significance of semantic connectivity in addressing these challenges. **Second**, we propose a Semantic Connectivity-driven pseudo-labeling (SeCo) algorithm, which can effectively generate high-quality pseudo-labels, thereby facilitating robust domain adaptation. **Third**, extensive experiments underscore the versatility of SeCo in effectively addressing various cross-domain semantic segmentation tasks, including domain generalization, traditional, even source-free [38, 28], and black-box [89] domain adaptation. Notably, SeCo achieves marked enhancements in the more challenging source-free and black-box domain adaptation tasks.

## 2 Method

**Problem Definition**. Cross-domain semantic segmentation aims to transfer a segmentor trained on the labeled source domain $D_s = \{(x_s^i, y_s^i)\}_{i=1}^{I_s}$ to the unlabeled target domain $D_t = \{(x_t^i)\}_{i=1}^{I_t}$, where $I_s$ and $I_t$ indicate the number of samples for each domain respectively. $x$ and $y$ represent an image and corresponding ground-truth label. Presently, mainstream cross-domain segmentation methods optimize the following objective to enhance model adaptability,

$$\mathcal{L} = \mathcal{L}_s(x_s, y_s) + \beta\mathcal{L}_t(x_t, \hat{y}_t), \tag{1}$$

where $\mathcal{L}_s$ is the supervised cross-entropy loss, $\beta$ is the trade-off weight, $\mathcal{L}_t$ is the unsupervised pseudo-labeling loss, and $\hat{y}_t$ is the pseudo-label. This formula underscores the critical importance of the quality of pseudo-labels in improving the model's cross-domain ability. To alleviate the noise in pseudo-labels, various estimation [97, 3] and calibration [73, 7, 42, 79] methods have been introduced for pseudo-label selection. However, as mentioned above, the filtered pseudo-labels still encounter issues of constrained semantics and challenging localization of category noise.

### 2.1 Overview

The presented **Se**mantic **Co**nnectivity-driven pseudo-labeling (SeCo) is illustrated in Fig. 2. SeCo comprises two components, namely Pixel Semantic Aggregation (PSA) and Semantic Connectivity Correction (SCC), working collaboratively to refine the low-quality and noisy pseudo-labels into high-quality and clean pseudo-labels. Initially, PSA aggregates pixels from the filtered pseudo-labels into connections by interacting with the *segment anything model* (SAM) [36] through stuff and things interactions. Subsequently, PSA segments the image into multiple connectivities based on their semantics. Guided by the connectivity set, SCC establishes a connectivity classifier, conducts

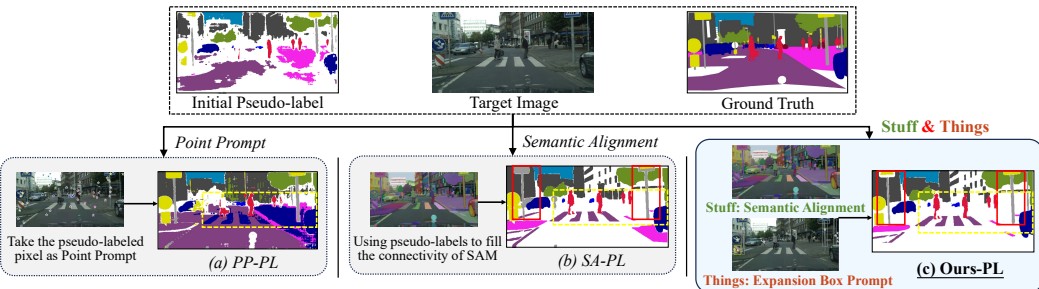

Figure 3: Comparison of Pseudo-Label (PL) aggregation using different interactive methods with SAM [36]. Both Point Prompt-based Interaction (PP-PL) and Semantic Alignment-based Interaction (SA-PL) amplify pseudo-label noise, whereas our method alleviates this issue.

connectivity pooling on image features, and classifies each connectivity. Leveraging information about fitting difficulty and loss distribution, SCC identifies and corrects noise. Finally, the connectivity-driven pseudo-labels, characterized by comprehensive semantics and low noise are achieved.

## 2.2 Pixel Semantic Aggregation

**Motivation: Why SAM?** Pixel semantic aggregation (PSA) proposes utilizing reliable pixels within pseudo-labels as category references and subsequently aggregating pixels that share similar semantics into connections. Intuitively, the above goals can be achieved through interactive segmentation [66, 44, 62] or pixel clustering [55, 1], but traditional techniques often struggle to accurately identify semantic boundaries in complex scenes, resulting in ambiguous aggregation. The advent interactive segmentation model, *segment anything model* (SAM) [36], provides powerful semantic capture capabilities. With reasonable prompts, SAM has the potential to give accurate semantic boundaries even in complex scenes [47, 33]. Building on SAM's remarkable capability, we are motivated to investigate how to leverage the reliable yet limited pseudo-labels to prompt SAM effectively and enhance the completion of pseudo-label semantics.

**Discussion on Utilization of SAM.** We analyze two naive solutions as outlined below. The first involves sampling the center pixels of each connected region on the pseudo-label as prompt points, as depicted in Fig. 3(a). We observe that when prompt points of the same category contain noise, this method compromises the aggregated segmentation structure. The disruption is attributed to noisy prompts interfering with the cross-attention mechanism in SAM [36].

The second method represents an improved way, called semantic alignment [5], aligning pseudo-labels with the connectivity established by SAM. This involves selecting the pseudo-label with the maximum proportion in each connectivity as the category for the entire region, as illustrated in Fig. 3 (b). We note that while this approach can refine pseudo-labels, it is consistently influenced by SAM's uncertain semantic granularity, particularly in the context of neighboring instance objects. Fig. 3 (b) provides examples of failures in this method, where SAM aggregates two categories, "traffic sign" and "pole" into a semantic connected region, leading to misaligned pseudo-labels due to this uncertainty. Our analysis indicates that this issue arises because SAM constructs connectivity by uniformly sampling points in space as prompts and subsequently filtering out redundantly connected regions. This fails to ensure corresponding sampling points for neighboring instance objects, resulting in a semantic granularity deviation between SAM's connectivity and specific segmentation tasks.

In summary, "prompts" interaction can aid in determining semantic granularity but is vulnerable to noise; Conversely, "alignment" interaction can alleviate noise interference but is susceptible to uncertain semantic granularity. **Hence, implementing SAM in cross-domain segmentation is a considerable challenge without a thoughtful design.** (For more discussion about SAM, see Sec.3.2)

**Our Strategy.** Building upon the analysis above, we find that noise significantly affects "stuff" categories due to their larger size and higher pixel proportions, making them more prone to selecting noisy pixels. On the other hand, semantic granularity uncertainty is more prevalent in "things" categories, given their smaller size and dense adjacency. To this end, we propose to interact with SAM in the form of "stuff" and "thing". Specifically, for "stuff", we utilize semantic alignment to mitigate the impact of noisy prompts, while for "things", we employ box and point prompts to guide the semantic precision. The detailed algorithm is in Algorithm 1. An illustration of the proposed strategy is shown in Fig. 3(c).

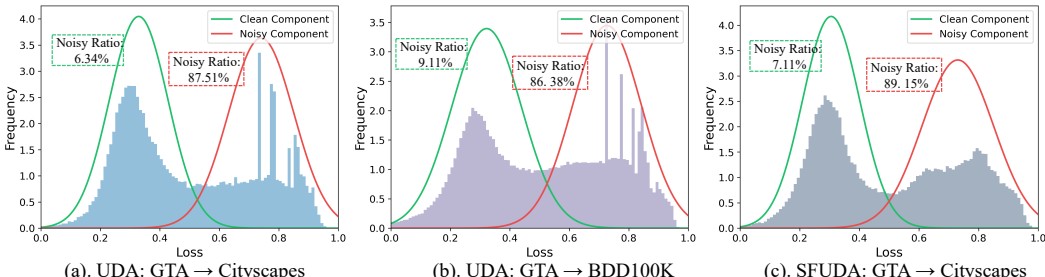

Figure 4: The loss distribution plot of semantic connectivity on different cross-domain segmentation tasks. By establishing a bi-modal Gaussian function, noisy connectivity can be effectively located.

## 2.3 Semantic Connectivity Correction

PSA aggregates both precise and noisy pseudo-labels into connectivities, facilitating the locating and correction of noise. This simplicity arises from the fact that distinguishing noise at the connectivity level is much easier than at the pixel level, as it eliminates the necessity to scrutinize local semantics and instead focuses on the overall category (See experimental analysis in Appendix D).

Inspired by this, we propose Semantic Connectivity Correction (SCC), introducing a simple connectivity classification task and detecting noise through loss distribution. Specifically, given the input image $x_i$, we first obtain the connectivity mask list $M = \{m^{i,n}\}_{n=1}^{N_i}$ and its corresponding connectivity-level pseudo-label $\hat{y}_{sc} = \{\hat{y}_{sc}^{i,n}\}_{n=1}^{N_i}$ from PSA, where $N_i$ represents the number of connectivities for the $i$-th sample $x_i$. Then, we set up a connectivity classifier, comprising a feature extractor $F_{scc}$ and a linear layer MLP, and optimize it with the following objective,

$$L_{scc} = \sum_{i,k,n} -\hat{y}_{sc}^{i,n,k} \log(\text{MLP}[\text{Pool}[F_{scc}(x^i), m^{i,n}])). \tag{2}$$

Pool$[\cdot, \text{mask}]$ denotes the average pooling of features corresponding to the input mask, $k \in [0, 1, ...K]$, and $K$ is the category number. Optimizing $L_{scc}$ conducts a $K$-way classification for each connectivity with clean and noisy labels.

Based on observations of *early learning* in noisy learning[88, 45, 82]: when training on noisy labels, deep neural networks, in the early stages of learning, initially match the training data with clean labels, and subsequently memorize examples with erroneous labels. We warm up the connectivity classifier for several epochs and then obtain the loss distribution by calculating Eq. (2) for each connectivity. As shown in Fig. 4, the loss of connectivities presents bimodol distribution, and the clusters with larger losses correspond to higher noise, which better conforms to the observations. To this end, we employ a two-component Gaussian Mixture Model to effectively model the loss distribution using the Expectation-Maximization algorithm [82]. Subsequently, the probability of connectivity being noisy, denoted as $\eta$, can be reasonably approximated by the Gaussian distribution associated with bigger loss, *i.e.*, $\eta^{i,n} = p(c|L_{scc}(x, m^{i,n}))$. $c$ is the parameters of the corresponding Gaussian distribution. We keep the clean connectivity by setting a noise threthod $\tau_{ns}$, *i.e.*,

$$D_{clean} = \{(x_i, y_{sc}^{i,n}) | \eta^{i,n} < \tau_{ns}\}. \tag{3}$$

Besides, we find that many noisy connectivities can be corrected by setting another correction threshold $\tau_{cr}$ on the output probability of connectivity classifier, *i.e.*,

$$D_{corr} = \{(x^i, k) | p_{scc}^{i,n,k} > \tau_{cr}, \eta^{i,n} > \tau_{ns}\}, \tag{4}$$

where $p_{scc}^{n,k}$ represents the probability of class $k$ for the $n$-th connectivity. We take the union of the two sets as the final connectivity-driven pseudo-label set $D_{all} = D_{clean} \cup D_{corr}$ for self-training.

## 2.4 Implementation on Domain Adaptation & Generalization Tasks

We provide solutions on connectivity-driven pseudo-labels for different domain adaptation tasks.

**Domain Generalization(DG).** Following [4], we first use Stable Diffusion to synthesize simulated unseen domain data. Then, we use the DG model and our SeCo to pseudo-label these synthesized data and retrain the DG model on them.

**Unsupervised Domain Adaptation.** The connectivity-driven pseudo-label set $D_{all}$ serves two primary functions. Firstly, they contribute to the pseudo-labeling $L_t$ loss in Eq. (1), providing

accurate semantic guidance for the target domain. The second objective is to mitigate category bias in domain adaptation. We treat $D_{all}$ as a sample pool, where we resample minority classes in the target domain and duplicate them through copy-paste operation [16] onto both domains.

**Source-free & Black-box Domain Adaptation.** In these scenarios, source access is restricted. This limitation prevents the deployment of the source loss $L_s$ in Eq. (1), making self-training more vulnerable to noise interference. Connectivity-driven pseudo-label set $D_{all}$ brings a novel idea to mitigate these challenges. With its contribution to accurate semantics and low noise, $D_{all}$ can be viewed as a well-organized labeled set, thereby transforming source-free and black-box domain adaptation tasks into semi-supervised segmentation tasks[84, 83].

**Discussion about Fair Comparison.** We acknowledge the potential concern regarding unfair comparisons between our SAM-incorporating method and existing approaches. **First**, it is important to emphasize the considerable challenges in applying SAM to CDSS tasks. Detailed experiments are in Table 5 of Section 3.2. We believe our work makes a significant contribution to exploring the potential of SAM-enhanced CDSS tasks. **Second**, our method is designed to be integrative, enhancing existing pseudo-labeling methods rather than competing with them (as demonstrated in Section 3.1 and Tables 1 and 2). **Third**, we conducted experiments without using SAM to validate that the proposed semantic connectivity denoising idea still has advantages, as shown in Fig. 5 of Section 3.2. We hope this work can inspire the community to further investigate the effective utilization of SAM in CDSS, embracing the popular trend of facilitating visual tasks with large-scale models, such as enhancing classification [52, 81, 61] with large-language models (*e.g.,* GPT-3).

# 3 Experiments

**Datasets.** We employ two real datasets (Cityscapes [12] and BDD-100k [86]) alongside two synthetic datasets (GTA5 [63] and SYNTHIA [65]). The details of these datasets are introduced in Section B.

**Implementation Details.** *Traditional UDA*: We opted for two network architectures: DeepLabV2 [6] with ResNet-101 [19] and SegFormer [80] with MiT-B5. For DeepLabV2, we chose two classical methods, AdvEnt [70] and ProDA [90], as the baselines. For SegFormer, we selected two highly successful UDA methods, DAFormer [24] and HRDA [25], as the baselines. *Source-free UDA*: We maintain DeepLabV2 as the base network to align with existing works. We chose HCL [28] and the current SOTA method DTST [93] as baselines. *Black-box UDA*: We use two SOTA black-box UDA methods, DINE [43] and BiMem [89], as baselines. Across all tasks, for each baseline method, we select the pseudo-labels of its predicted top 50% confidence-ranked pixels for SeCo processing. Subsequently, we utilize the SAM [36] with Vision Transformer-H (ViT-H) [14] to generate connectivities. We refrain from using SAM to refine pseudo-labels for test data to avoid introducing extra inference overhead. The automatic mask generation process in SAM adheres to the official parameter settings. In Algorithm 1, the enlargement factor for the bounding box area is set to 1.5. The connectivity classifier is trained only for 5000 iterations in an early learning way for all tasks. The noise threshold ($\tau_{ns}$) and correction threshold ($\tau_{cr}$) are configured at 0.60 and 0.95, respectively.

## 3.1 Combined SeCO with State-of-the-Art (SOTA) Methods

The performance of SeCo is shown in Tables 1, 2, 3 and 4. Overall, experimental results indicate that SeCo can be integrated with various SOTA domain adaptation and domain generalization methods, and significantly enhances their adaptability. Moreover, SeCo exhibits notable improvements for both source-free and black-box adaptation, overcoming limitations with the source domain data.

**Domain Generalization**. In this experiment, we followed CLOUDS [4], which uses synthetic data from Stable Diffusion (SD) to assist DG. Our aims are: 1) to show that our method can handle challenging synthesized data with open-set noise, and 2) to compare our SAM usage with the competitive scheme in CLOUDS. Compared to the DG baseline SHADE and HRDA, our method significantly improves their performance by 6.8% and 3.1%, respectively. Compared to CLOUDS, which also uses SAM, we achieve even greater improvements, further enhancing performance by 1.4% and 1.0%.

**Domain Adaptation**. *GTA5 → Cityscapes*. Results are reported in Table 1. In the UDA setting, the integration of SeCo with AdvEnt [70] leads to a notable performance improvement, achieving a 13.4% increase in mIoU score. Combining SeCo with ProDA [90] results in a 9.4% increase in

| | Road | S.walk | Build. | Wall | Fence | Pole | Tr.Light | sign | Veget. | terrain | Sky | Person | Rider | Car | Truck | Bus | Train | M.bike | Bike | mIoU |
|---|---|---|---|---|---|---|---|---|---|---|---|---|---|---|---|---|---|---|---|---|
| **Unspervised domain adaptation: GTA → Cityscapes** | | | | | | | | | | | | | | | | | | | | |
| AdvEnt [70] ICCV'19 | 89.4 | 33.1 | 81.0 | 26.6 | 26.8 | 27.2 | 33.5 | 24.7 | 83.9 | 36.7 | 78.8 | 58.7 | 30.5 | 84.8 | 38.5 | 44.5 | 1.7 | 31.6 | 32.4 | 45.5 |
| AdvEnt + Ours | 92.0 | 61.0 | 87.0 | 51.0 | 49.4 | 48.9 | 44.5 | 44.3 | 86.7 | 50.0 | 87.9 | 63.3 | 46.0 | 89.7 | 57.6 | 54.6 | 5.6 | 47.7 | 51.6 | **58.9** (+13.4) |
| ProDA [90] CVPR'21 | 91.5 | 52.4 | 82.9 | 42.0 | 35.7 | 40.0 | 44.4 | 43.3 | 87.0 | 43.8 | 79.5 | 66.5 | 31.4 | 86.7 | 41.1 | 52.5 | 0.0 | 45.4 | 53.8 | 53.7 |
| ProDA + Ours | 94.4 | 65.6 | 87.8 | 55.8 | 54.7 | 56.8 | 58.6 | 60.3 | 90.2 | 51.5 | 93.7 | 72.7 | 48.0 | 88.1 | 51.3 | 65.3 | 1.5 | 60.3 | 61.0 | **64.1** (+9.4) |
| DAFormer [24] CVPR'22 | 95.7 | 70.2 | 89.4 | 53.5 | 48.1 | 49.6 | 55.8 | 59.4 | 89.9 | 47.9 | 92.5 | 72.2 | 44.7 | 92.3 | 74.5 | 78.2 | 65.1 | 55.9 | 61.8 | 68.2 |
| DAFormer+Ours | 96.2 | 74.4 | 90.9 | 56.7 | 49.7 | 60.5 | 62.7 | 69.4 | 92.4 | 54.9 | 93.9 | 77.1 | 53.1 | 96.6 | 83.1 | 82.2 | 72.5 | 62.6 | 65.6 | **73.4** (+5.3) |
| HRDA [25] ECCV'22 | 96.4 | 74.4 | 91.0 | 61.6 | 51.5 | 57.1 | 63.9 | 69.3 | 91.3 | 48.4 | 94.2 | 79.0 | 52.9 | 93.9 | 84.1 | 85.7 | 75.9 | 63.9 | 67.5 | 73.8 |
| HRDA+Ours | 96.6 | 80.9 | 92.4 | 62.5 | 57.5 | 61.0 | 66.7 | 71.7 | 92.4 | 52.3 | 95.1 | 80.6 | 56.3 | 95.9 | 86.1 | 86.6 | 76.8 | 65.4 | 68.7 | **76.1** (+2.3) |
| **Source-free domain adaptation: GTA → Cityscapes** | | | | | | | | | | | | | | | | | | | | |
| HCL [28] NIPS'21 | 92.0 | 55.0 | 80.4 | 33.5 | 24.6 | 37.1 | 35.1 | 28.8 | 83.0 | 37.6 | 82.3 | 59.4 | 27.6 | 83.6 | 32.3 | 36.6 | 14.1 | 28.7 | 43.0 | 48.1 |
| HCL+Ours | 94.6 | 62.5 | 88.6 | 48.4 | 41.6 | 45.2 | 43.5 | 32.9 | 84.0 | 45.3 | 91.6 | 66.0 | 47.5 | 89.0 | 42.6 | 58.8 | 31.5 | 47.2 | 56.2 | **58.8** (+10.6) |
| DTST [93] CVPR'23 | 90.3 | 47.8 | 84.3 | 38.8 | 22.7 | 32.4 | 41.8 | 41.2 | 85.8 | 42.5 | 87.8 | 62.6 | 37.0 | 82.5 | 25.8 | 32.0 | 29.8 | 48.0 | 56.9 | 52.1 |
| DTST+Ours | 94.9 | 65.9 | 89.9 | 48.2 | 42.3 | 45.9 | 48.9 | 45.6 | 85.7 | 46.2 | 91.1 | 68.2 | 47.6 | 88.5 | 44.9 | 57.8 | 29.5 | 50.7 | 57.8 | **60.5** (+8.4) |
| **Black-box domain adaptation: GTA → Cityscapes** | | | | | | | | | | | | | | | | | | | | |
| DINE [43] CVPR'22 | 88.2 | 44.2 | 83.5 | 14.1 | 32.4 | 23.5 | 24.6 | 36.8 | 85.4 | 38.3 | 85.3 | 59.8 | 27.4 | 84.7 | 30.1 | 42.2 | 0.0 | 42.7 | 45.3 | 46.7 |
| DINE+Ours | 89.6 | 60.8 | 84.1 | 46.3 | 38.4 | 44.0 | 41.6 | 32.2 | 82.1 | 41.7 | 86.6 | 63.4 | 44.9 | 83.9 | 41.5 | 58.6 | 0.0 | 40.5 | 54.1 | **54.4** (+7.7) |
| BiMem [89] ICCV'23 | 94.2 | 59.5 | 81.7 | 35.2 | 22.9 | 21.6 | 10.0 | 34.3 | 85.2 | 42.4 | 85.0 | 56.8 | 26.4 | 85.6 | 37.2 | 47.4 | 0.2 | 39.9 | 50.9 | 48.2 |
| BiMem+Ours | 93.9 | 61.4 | 87.6 | 47.7 | 41.3 | 44.0 | 43.2 | 32.7 | 83.2 | 44.4 | 91.4 | 66.9 | 46.6 | 88.7 | 42.6 | 60.8 | 0.0 | 46.2 | 55.0 | **56.7** (+8.5) |

Table 1: Performance improvement in terms of mIoU score (%) by incorporating SeCo into existing DA methods, where GTA5 serves as the source domain.

| | Road | S.walk | Build. | Wall | Fence | Pole | Tr.Light | sign | Veget. | Sky | Person | Rider | Car | Bus | M.bike | Bike | mIoU |
|---|---|---|---|---|---|---|---|---|---|---|---|---|---|---|---|---|---|
| **Unsupervised domain adaptation: SYNTHIA → Cityscapes** | | | | | | | | | | | | | | | | | |
| AdvEnt [70] ICCV'19 | 87.0 | 44.1 | 79.7 | 9.6 | 0.6 | 24.3 | 4.8 | 7.2 | 80.1 | 83.6 | 56.4 | 23.7 | 72.7 | 32.6 | 12.8 | 33.7 | 40.8 |
| AdvEnt + Ours | 87.9 | 47.7 | 82.9 | 20.1 | 1.1 | 38.2 | 29.2 | 28.6 | 86.5 | 85.7 | 64.5 | 29.6 | 84.5 | 44.3 | 39.1 | 47.4 | **51.1** (+10.3) |
| ProDA [90] CVPR'21 | 87.1 | 44.0 | 83.2 | 26.9 | 0.7 | 42.0 | 45.8 | 34.2 | 86.7 | 81.3 | 68.4 | 22.1 | 87.7 | 50.0 | 31.4 | 38.6 | 51.9 |
| ProDA + Ours | 88.1 | 49.8 | 86.9 | 33.9 | 1.4 | 46.6 | 54.3 | 44.7 | 85.8 | 85.7 | 84.1 | 40.3 | 86.0 | 55.2 | 45.0 | 50.6 | **58.6** (+6.7) |
| DAFormer [24] CVPR'22 | 84.5 | 40.7 | 88.4 | 41.5 | 6.5 | 50.0 | 55.0 | 54.6 | 86.0 | 89.8 | 73.2 | 48.2 | 87.2 | 53.2 | 53.9 | 61.7 | 60.9 |
| DAFormer + Ours | 88.9 | 49.9 | 90.7 | 46.2 | 7.3 | 55.0 | 63.2 | 57.8 | 87.7 | 92.7 | 76.0 | 51.5 | 89.5 | 61.3 | 59.7 | 64.9 | **65.1** (+4.2) |
| HRDA [25] ECCV'22 | 85.2 | 47.7 | 88.8 | 49.5 | 4.8 | 57.2 | 65.7 | 60.9 | 85.3 | 92.9 | 79.4 | 52.8 | 89.0 | 64.7 | 63.9 | 64.9 | 65.8 |
| HRDA + Ours | 90.7 | 50.6 | 89.8 | 51.6 | 8.4 | 59.4 | 66.9 | 64.9 | 89.1 | 95.5 | 81.9 | 58.2 | 91.4 | 66.3 | 65.4 | 66.1 | **68.5** (+2.3) |
| **Source-free domain adaptation: SYNTHIA → Cityscapes** | | | | | | | | | | | | | | | | | |
| HCL [28] NIPS'21 | 80.9 | 34.9 | 76.7 | 6.6 | 0.2 | 36.1 | 20.1 | 28.2 | 79.1 | 83.1 | 55.6 | 25.6 | 78.8 | 32.7 | 24.1 | 32.7 | 43.5 |
| HCL + Ours | 88.3 | 46.0 | 83.3 | 10.6 | 1.5 | 38.6 | 29.3 | 29.0 | 86.9 | 86.0 | 64.6 | 30.0 | 84.7 | 44.7 | 39.2 | 47.7 | **50.7** (+7.2) |
| DTST [93] CVPR'23 | 79.4 | 41.4 | 73.9 | 5.9 | 1.5 | 30.6 | 35.3 | 19.8 | 86.0 | 86.0 | 63.8 | 28.6 | 86.3 | 36.6 | 35.2 | 53.2 | 47.7 |
| DTST + Ours | 88.7 | 48.5 | 87.4 | 23.5 | 2.3 | 39.2 | 30.3 | 31.9 | 91.1 | 86.8 | 64.7 | 33.4 | 88.6 | 45.1 | 43.3 | 57.9 | **53.9** (+6.2) |
| **Black-box domain adaptation: SYNTHIA → Cityscapes** | | | | | | | | | | | | | | | | | |
| DINE [43] CVPR'22 | 77.5 | 29.6 | 79.5 | 4.3 | 0.3 | 39.0 | 21.3 | 13.9 | 81.8 | 68.9 | 66.6 | 13.9 | 71.7 | 33.9 | 34.2 | 18.6 | 40.9 |
| DINE + Ours | 86.7 | 43.9 | 82.1 | 6.8 | 0.0 | 32.5 | 28.3 | 26.7 | 82.1 | 83.9 | 60.0 | 25.1 | 79.1 | 39.8 | 36.5 | 45.8 | **47.5** (+6.6) |
| BiMem [89] ICCV'23 | 78.8 | 30.5 | 80.4 | 5.9 | 0.1 | 39.2 | 21.6 | 15.0 | 84.7 | 74.3 | 66.8 | 14.1 | 73.3 | 36.0 | 32.3 | 21.8 | 42.2 |
| BiMem + Ours | 84.5 | 43.8 | 79.2 | 8.1 | 0.9 | 39.8 | 25.3 | 25.6 | 85.7 | 85.1 | 63.4 | 29.7 | 82.8 | 40.9 | 35.9 | 44.2 | **48.4** (+6.2) |

Table 2: Performance improvement in terms of mIoU score (%) by incorporating SeCo into existing DA methods, where SYNTHIA serves as the source domain.

mIoU score, establishing a new SOTA using DeepLabV2. When integrated with the high-performing Segformer [80], SeCo consistently improves DAFormer by 5.3% in mIoU score and HRDA by 2.3% in mIoU score. In source-free UDA, SeCo exhibits stronger advantages, providing robust self-training, and elevating the performance of existing SOTA methods, HCL and DTST, by 10.6% and 8.5%, respectively. In the more stringent black-box adaptation setting, SeCo remains effective. When integrated with two SOTA methods, DINE and BiMem, SeCo obtains improvements by 7.7% and 8.5%, respectively. *SYNTHIA → Cityscapes*. Results are reported in Table 2. Despite the larger domain shift of this task, SeCo maintains similar improvements as the previous task, which further underscores the potential of SeCo under data protection scenarios. *GTA5 → BDD-100k*. Results are reported in Table 4. This task involves complex mixed-weather adaptation. SeCo consistently achieves stable performance improvements. Specifically, SeCo enhances the performance of two baseline methods, PyCDA [46] and ProDA [90], by 11.4% and 7.8%, respectively, establishing itself as the new state of the art for this benchmark. In the source-free setting, SeCo achieves an improvement of 6.2% over the state-of-the-art method [95], demonstrating sustained and stable performance gains.

## 3.2 Analysis and Ablation Study

**Can SAM benefit Cross-Domain Semantic Segmentation in another naive way?** In Table 5, we conduct three types of experiments to demonstrate that *directly applying SAM on Cross-*

| | Backbone | Using SAM | Cityscapes | BDD-100K | Mapillary | Average |
|---|---|---|---|---|---|---|
| SHADE [96] IJCV'23 | | ✗ | 46.6 | 43.7 | 45.5 | 45.3 |
| TLDR [35] ICCV'23 | | ✗ | 47.6 | 44.9 | 48.8 | 47.1 |
| MoDify [32] ICCV'23 | ResNet-101 | ✗ | 48.8 | 44.2 | 47.5 | 46.8 |
| + CLOUDS [4] CVPR'24 | | ✓ | 50.6 | 44.8 | 56.6 | 50.7 |
| + SeCo (Ours) | | ✓ | **52.4** | **46.1** | **57.7** | **52.1** |
| HRDA [25] ECCV'22 | | ✗ | 57.4 | 49.1 | 61.1 | 55.9 |
| + CLOUDS [4] CVPR'24 | MiT-B5 | ✓ | 58.1 | 53.8 | 62.3 | 58.1 |
| + SeCo (Ours) | | ✓ | **58.8** | **54.9** | **63.6** | **59.1** |

Table 3: Performance improvement in terms of mIoU score (%) by incorporating SeCo into existing domain generalization methods using GTA5 as the source domain.

| SourceGTA5→ | Compound | | | Open | Average |
| | Rainy | Snowy | Cloudy | Overcast | Compound + Open Overcast |
|---|---|---|---|---|---|
| Source Only | 28.7 | 29.1 | 33.1 | 32.5 | 30.9 |
| Unsupervised domain adaptation: GTA5 → BDD-100k | | | | | |
| ML-BPM [56] ECCV'22 | 40.5 | 39.9 | 42.1 | 40.9 | 40.9 |
| OSC [15] NIPS'23 | - | - | - | - | 44.0 |
| PyCDA [46] CVPR'20 | 33.4 | 32.5 | 36.7 | 37.8 | 35.1 |
| PyCDA + Ours | 43.6 | 42.1 | 49.7 | 50.7 | **46.5** (+11.4) |
| ProDA [90] CVPR'21 | 40.3 | 40.6 | 43.2 | 42.5 | 41.7 |
| ProDA + Ours | 47.6 | 45.7 | 51.9 | 52.6 | **49.5** (+7.8) |
| Source-free domain adaptation: GTA5 → BDD-100k | | | | | |
| SFOCDA [95] TCSVT'22 | 35.4 | 33.4 | 41.4 | 41.2 | 37.9 |
| SFOCDA + Ours | 41.7 | 42.1 | 44.7 | 47.9 | **44.1** (+6.2) |

Table 4: The comparison of performance in terms of mIoU score (%) on the Open Compoud domain adaptation task between SeCo (ours) and other state-of-the-art methods.

*Domain Semantic Segmentation (CDSS) can hardly obtain improvement.* ① Use the backbone of SAM to empower CDSS. Table 5 shows that **utilizing SAM as the backbone achieves a notable performance drop** in UDA. The reduction mainly came from rare classes (*e.g.* "train" and "truck"). Both DAFomer and HRDA use Feature Distance to keep the semantic knowledge of ViT-B pre-trained on ImageNet, effectively improving the adaptation for rare classes. However, SAM's pre-training does not consider such semantic knowledge and thus obtains inferior results in rare classes. ② Is the UDA still necessary, if we use CLIP+SAM (CSAM) to get the initial pseudo label? We conduct the following experiment to explore the feasibility of CLIP+SAM: a) Use SAM to segment the input image. b) Extract the largest bounding rectangle from each segment. c) Create text descriptions for categories, e.g., "a photo of a road." d) Use CLIP to match image patches with text descriptions, assigning text labels as categories. CLIP+SAM+UDA combines pseudo-labels from CLIP+SAM and UDA, using voting fusion to select consistent predictions for training. Table 5 shows that **CSAM cannot obtain competitive results** on the target data, even when combined with UDA methods. The reasons are below. Given spatially uniform sampling as prompt points, 1) SAM is prone to over-segmentation results, which makes it difficult for CLIP to obtain sufficient context from small segments. 2) SAM's segments may conflict with the defined semantics of the target data, *e.g.*, SAM always treats 'poles' and 'traffic sign' as one segment. ③ Is using SAM on coarse pseudo-labels enough? Table 5 shows the comparison of using vanilla SAM and our method on the source-trained model's (Coarse) and UDA-adapted model's prediction. It shows that **the gains of using SAM on coarse pseudo-labels are minimal** and even negative on strong UDA baselines. This is because SAM risks introducing more semantic noise when extending semantic boundaries. Our method, even with coarse pseudo-labels, allows SAM to achieve greater benefits. The above discussions indicate the difficulty of applying SAM in cross-domain segmentation and the non-trivial design of our SAM-based method.

**Is Our Method Specific to SAM?** The proposed Semantic Connectivity Correction (SCC) is general and not specific to SAM. SCC can work on any form of pseudo-labels, although their connectivity structure may not be as good as SAM's. In Fig. 5, we report the results of directly using SCC on pseudo-labels generated by existing UDA models. Under this exact fair comparison, our method still achieves good improvement and is more competitive than widely used Knowledge Distillation (KD) [90]. Again, we want to emphasize that this work aims to bring a new perspective for enhancing CDSS to embrace the huge benefit of large-scale models, which is a non-trivial contribution.

| ① Use the backbone of SAM to empower CDSS | | | | | | |
|---|---|---|---|---|---|---|
| | DAFormer | + (SAM ViT-B) | + Ours | HRDA | + (SAM ViT-B) | + Ours |
| **G2C** | 68.2 | 64.1 (-4.1) | **73.4 (+5.2)** | 73.8 | 69.1 (-4.7) | **76.1 (+2.3)** |
| **S2C** | 60.9 | 60.2 (-0.7) | **65.1 (+4.2)** | 65.8 | 63.7 (-2.1) | **68.5 (+2.3)** |
| ② Use CLIP + SAM (CSAM) to get the initial pseudo-label | | | | | | |
| | CSAM | CSAM + DAFormer | Ours + DAFormer | CSAM + HRDA | Ours + HRDA | |
| **G2C** | 43.7 | 69.1 | **73.4** | 73.5 | **76.1** | |
| **S2C** | 41.7 | 61.1 | **65.1** | 65.2 | **68.5** | |

| ③ Use SAM on coarse pseudo-labels | | | | | | |
|---|---|---|---|---|---|---|
| Method | Initial | Using Source-model's PL | | Using UDA's PL | | Upper Bound |
| | | Vanilla SAM | Ours | Vanilla | Ours | |
| AdvEnt [70] ICCV'19 | 45.5 | 48.6 (+3.1) | 53.9 (+8.4) | 50.9 (+5.5) | **58.9 (+13.4)** | 69.1 |
| ProDA [90] CVPR'21 | 53.7 | 50.1 (-1.7) | 58.2 (+4.5) | 57.9 (+4.3) | **64.1 (+9.4)** | 69.1 |
| DAFormer [24] CVPR'22 | 68.2 | 67.7 (-0.5) | 69.9 (+1.7) | 69.7 (+1.5) | **73.4 (+5.3)** | 76.4 |
| HRDA [25] ECCV'22 | 73.8 | 72.7 (-1.1) | 74.6 (+0.8) | 74.6 (+0.8) | **76.1 (+2.3)** | 77.1 |

Table 5: Comparison of different ways of applying SAM to cross-domain semantic segmentation (CDSS). G2C is GTA5 → Cityscape. S2C is SYNTHIA → Cityscape. ③ is carried on G2C.

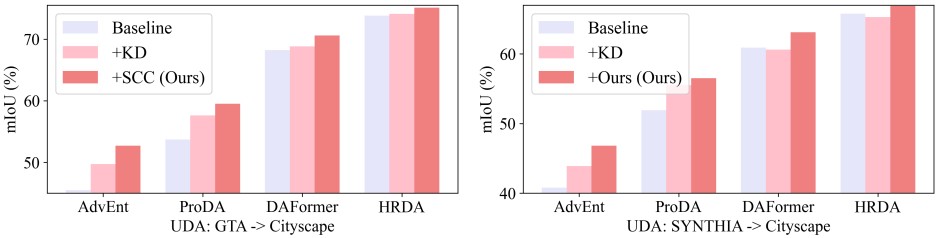

Figure 5: Comparison between widely used pixel-level distillation [90] and Semantic Connectivity Correction (SCC) without using SAM across various baselines.

| Baselines | Settings | PSA$^{(b)}$ | PSA | SCC | mIoU | Baselines | Settings | PSA$^{(b)}$ | PSA | SCC | mIoU |
|---|---|---|---|---|---|---|---|---|---|---|---|
| ProDA [90] CVPR'21 | UDA | ✓ | | | 53.7 | DTST [93] CVPR'23 | SF-UDA | ✓ | | | 52.1 |
| | | | ✓ | | 60.9 | | | | ✓ | | 56.1 |
| | | | ✓ | | 62.1 | | | | ✓ | | 57.9 |
| | | | ✓ | ✓ | **64.1** | | | | ✓ | ✓ | **60.5** |
| DAFormer [24] CVPR'22 | UDA | | | | 68.2 | BiMem [89] ICCV'23 | BB-UDA | ✓ | | | 48.2 |
| | | | ✓ | | 69.7 | | | | ✓ | | 52.2 |
| | | | ✓ | | 70.3 | | | | ✓ | | 54.4 |
| | | | ✓ | ✓ | **73.4** | | | | ✓ | ✓ | **56.7** |

Table 6: Ablation experiments of SeCo under various UDA settings on GTA5 → Cityscape adaptation task. PSA: Pixel Semantic Aggregation. SCC: Semantic Connectivity Correction. PSA$^{(b)}$ refers to the interaction with SAM using semantic alignment [5], as shown in Fig. 3. SF-UDA: source-free UDA. BB-UDA: black-box UDA.

**Ablation Study.** The results of ablation experiments are presented in Table 6. We conduct analyses across different domain adaptation settings as follows.

*In UDA*, PSA yields a performance improvement of 5.4% for ProDA [90], surpassing the gains achieved by PSA$^{(b)}$. Additionally, SCC builds upon PSA, contributing an additional 5.0% enhancement to ProDA. A similar trend is observed in the ablation study on DAFormer [24]. These findings suggest that, at the interaction level with SAM, PSA proves more effective than PSA$^{(b)}$; however, interacting solely with SAM is insufficient for achieving substantial self-training performance gains. SCC plays a crucial role in further filtering out noise propagated by SAM, leading to a significant enhancement in UDA performance.

*In source-free UDA*, PSA results in a 3.0% performance improvement for DTST [93], still outperforming PSA$^{(b)}$. Due to the substantial initial pseudo-label noise in the source-free setting, PSA aggregates more noisy connections, resulting in a diminished performance gain compared to UDA. SCC, building upon PSA, brings an improvement of 5.4%, reinforcing the notion that SCC can effectively filter and correct propagated pseudo-labels.

*In balck-box UDA*, PSA brings a marginal improvement, with only a gain of 1.2%. SCC on top of PSA achieves a substantial improvement of 7.3%, further confirming the aforementioned conclusions. These results underscore the importance of correcting noise within connections, especially under more significant domain shifts and weaker initial segmentation results.

| Prompt Way | Base (w/o SAM) | Prompting Only | Semantic Alignment | PSA | PSA+SCC |
|---|---|---|---|---|---|
| SeCo+ProDA (UDA) | 53.7 | 48.0 (-5.7) | 60.9 (+7.2) | **62.1 (+8.4)** | 64.1 |
| SeCo+DAFormer (UDA) | 68.2 | 64.6 (-3.6) | 69.7 (+1.5) | **70.3 (+2.1)** | 73.4 |
| SeCo+DTST (SF-UDA) | 52.1 | 46.7 (-5.4) | 56.1 (+4.0) | **57.9 (+5.8)** | 60.5 |
| SeCo+BiMem(BB-UDA) | 48.2 | 42.8 (-5.4) | 52.4 (+4.2) | **54.4 (+6.2)** | 56.7 |

Table 7: Ablation studies on "Prompting Only" (PO) and "Semantic Alignment"(SA) across multiple tasks in GTA → Cityscape.

| | GTA → Cityscapes (UDA) | SYNTHIA → Cityscapes (UDA) | GTA5 → BDD-100k (OC-DA) |
|---|---|---|---|
| ProDA (CVPR'21) | 53.7 | 51.9 | 41.7 |
| DivideMix [41] | 49.8 | 47.6 | 37.4 |
| PSA+DivideMix[41] | 60.1 | 53.4 | 44.2 |
| SeCo | **64.1** | **58.6** | **49.5** |

Table 8: Detailed comparison of our SCC and Dividemix across multiple domain adaptation tasks.

**Detailed ablation on prompt way.** We conduct ablation studies on "Prompting Only" (PO) and "Semantic Alignment"(SA) across multiple tasks in GTA → Cityscape in Table 7. We provide two metrics for these detailed ablations: PL mIoU (pseudo-label quality on the training set) and Val. mIoU (model performance on the validation set after training with those pseudo-labels). As shown in the Table 7, the "prompting only" method reduces the quality of pseudo-labels in the training set, leading to poor adaptation performance. This is because the unreliable interaction method introduces excessive noise into the pseudo-labels generated by SAM. "Semantic alignment" improves the quality of the training set pseudo-labels, but the improvement is limited, resulting in limited adaptation benefits. In contrast, our method enhances the quality of the training set pseudo-labels through better interaction, leading to superior performance gains.

**Ablation studies on our SCC and Dividemix.** Our SCC is partly inspired by DivideMix[41], however, we focus on mitigating the pixel-level noises in pseudo-labels raised by domain shifts and SAM refinement, while Dividemix focuses on mitigating the image-level label noises. Besides, we would like to emphasize that one of the main contributions of SCC is to provide the idea of denoising at the connectivity level, which makes it possible to apply other image-level denoising methods such as Dividemix to segmentation tasks. To better verify the effectiveness of our SCC, we make two experiments as show in Table 8: a) directly applying DivideMix to pixel-level denoising. b) using DivideMix to denoise the pixels aggregated by our PSA (using SAM). The results show that pixel-level denoising methods based on DivideMix are inferior to SCC even with SAM, highlighting the advantage of denoising at the connectivity level.

## 4 Conclusion

In this work, we propose Semantic Connectivity-driven Pseudo-labeling (SeCo), formulating pseudo-labels at the connectivity level for structured and low-noise semantics. SeCo, comprising Pixel Semantic Aggregation (PSA) and Semantic Connectivity Correction (SCC), efficiently aggregates speckled pseudo-labels into semantic connectivity with SAM. SCC introduces a simple connectivity classification task for locating and correcting connected noise. Experiments demonstrate SeCo's flexibility and significant effectiveness in performance across various cross-domain semantic segmentation tasks. We hope that this work could inspire the community to apply SAM to more cross-domain, semi-supervised and few-shot segmentation settings.

## 5 Acknowledgments

This work was supported by the National Natural Science Foundation of China under Grant Nos. 62271377, the National Key Research and Development Program of China under Grant Nos. 2021ZD0110400, 2021ZD0110404, the Key Research and Development Program of Shannxi (Program Nos. 2021ZDLGY01-06, 2022ZDLGY01-12, 2023YBGY244, 2023QCYLL28, 2024GX-ZDCYL-02-08, 2024GX-ZDCYL-02-17), the Key Scientific Technological Innovation Research Project by Ministry of Education, the State Key Program and the Foundation for Innovative Research Groups of the National Natural Science Foundation of China (61836009), the Joint Funds of the National Natural Science Foundation of China (U22B2054), the MUR PNRR project FAIR (PE00000013) funded by the NextGenerationEU, and the EU Horizon project ELIAS (No. 101120237).

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

# Appendix

## A   Related Work

**Domain Adaptive Semantic Segmentation (DASS)** transfers the source knowledge to the target mainly through the following avenues: *Source Domain Augmentation*: This approach involves employing style augmentation [85, 27, 96, 40] and domain randomization [17, 87, 34, 29, 20] to expand the representation space learned by the source domain model with limited data, thereby enhancing the model's generalization capability. *Minority Class Enhancement*: This line of work introduces minority class resampling [68, 16, 93, 49], minority class perturbation [75, 50], and minority class feature alignment [48] to enhance the adaptation capability of minority classes. *Aligning Source and Target Domains:* This line of work employ various domain alignment strategies, *e.g.*, adversarial training [23, 72], statistical matching [76], across diverse alignment spaces (*e.g.*, input [21, 71], feature [72] and output space [69]) to reduce statistical differences between the two domains. *Self-Training Techniques*: This line of methods primarily employs pseudo-labeling techniques to further address the issue of inadequate target adaptation. To counter pseudo-label noise, existing approaches employ various strategies, including introducing strong augmentations from input data [26], designing teacher-student model structures [25], and employing pseudo-label selection methods [3, 42, 57, 98, 99, 51, 94, 74], alleviating the issue of error accumulation. Our work is the first to formulate pseudo-labels at the connectivity level, thereby facilitating the learning of structured and low-noise semantics.

**Domain Generalizable Semantic Segmentation (DGSS)** is proposed to address the generalization problem to unseen domains, which is more realistic as we often cannot obtain target data in advance. In the computer vision field, existing DGSS methods usually regard style information as a domain factor and remove it or augment it explicitly to achieve generalization. For example, some advanced methods remove style-related factors by specific normalization [58] or whitening [11] operations. Another type of method attempts to expand the generalization boundary of the model by expanding diverse style data in global [87, 27] or class level [59]. However, style augmentation alone in the domain fails to enable the model to cover more unseen scenarios. Cloud [4] proposes leveraging the pre-trained Stable Diffusion model to simulate synthetic scenarios and uses SAM to complete their pseudo-labels, which further significantly improves the generalization of the model. Following this approach, our method can further filter out the label noise caused by utilizing SAM in the cloud.

**Segment Anything Model** (SAM) [36] has gained widespread attention, with multiple works incorporating it into specific segmentation tasks. For instance, [47] fine-tuned SAM in the medical domain to establish a robust foundational medical model. In few-shot learning, [91] applied SAM, achieving notable results with minimal parameter fine-tuning. [77] proposed an efficient method for fine-tuning SAM in downstream segmentation scenarios. Moreover, [5] combined SAM with semantic segmentation models to enhance segmentation model boundaries. Our approach signals the strong potential of SAM in pseudo-label-based cross-domain segmentation tasks. While [5] is closely related, we conduct a detailed analysis of its limitations in Section 2.2.

**Noisy Label Learning** (NLL). Currently, NLL focuses on classification tasks with techniques like robust loss design [92], regularization [78], label weighting [60], and correction [41]. These methods typically target image-level noise and may not be effective for pixel-level segmentation, which involves complex spatial and semantic dependencies among pixels. Maintaining spatial consistency across millions of pixels is a challenge for current image-level denoising methods. In segmentation, few methods focus on denoising the pseudo-label, such as ProDA and RPL, which denoise each pixel independently and still face the challenges highlighted in our paper. Our SeCo effectively links image-level techniques with segmentation tasks, offering novel solutions for pseudo-label denoising in segmentation.

## B   Dataset Details

We employ two real datasets (Cityscapes [12] and BDD-100k [86]) alongside two synthetic datasets (GTA5 [63] and SYNTHIA [65]). The Cityscapes dataset comprises 2,975 training images and 500 validation images, all with a resolution of 2048×1024. BDD-100k is a real-world dataset compiled from various locations in the United States. It encompasses a variety of scene images, including those captured under different weather conditions such as rain, snow, and clouds, all with a resolution of

**Algorithm 1** Aggregation of Pseudo-Labels with SAM

---

1: **procedure** AGGREGATEPSEUDOLABELS(Image $x$, Pseudo-Label $\hat{y}$, SAM Model)
2:     **Aggregate Pseudo-Labels for "things" category:**
3:     Extract connectivities of $\hat{y}$
4:     **for each connectivity** in $\hat{y}$ of "things" category **do**
5:         Compute enlarged maximum bounding box as box prompt
6:         Compute geometric center as point prompt
7:         Interact with SAM using box and point prompts
8:         Obtain aggregated connectivity
9:     **Aggregate Pseudo-Labels for "stuff" category:**
10:    Input $x$ into SAM to get no-semantic connectivities
11:    **for each "stuff" category** in $\hat{y}$ **do**
12:        Align $\hat{y}$ with the no-semantic connectivities by assigning the maximum proportion pseudo-label
13:        Obtain aggregated connectivities
14:    **Merge stuff and thing:**
15:    $\hat{y}_{\text{psa}} \leftarrow$ Merge and filter overlapping connectivities
16:    **Output:** Aggregated Pseudo-Label $\hat{y}_{\text{psa}}$

---

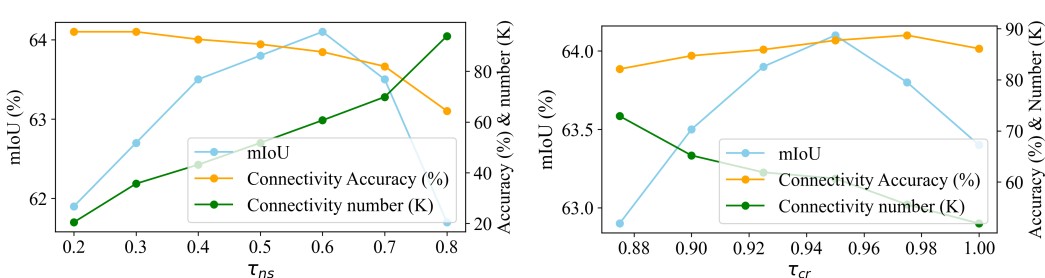

Figure 6: Evaluation on $\tau_{ns}$ and $\tau_{cr}$ in GTA5 → Cityscapes using ProDA [90] as baseline.

1280×720. The GTA5 dataset consists of 24,966 images with a resolution of 1914×1052, sharing 19 common categories with Cityscapes. The SYNTHIA dataset encompasses 9,400 images with a resolution of 1280×760, featuring 16 common categories with Cityscapes.

## C   Algorithm

We provide the procedure of aggregation of pseudo-labels in Algorithm 1.

## D   Futher Results

**Hyper-Parameter Sensitivity.** Fig. 6 illustrates the impact of $\tau_{ns}$ and $\tau_{cr}$ on the final model's adaptability (mIoU), connectivity accuracy, and the kept number of connectivity. We set the range of $\tau_{ns}$ from 0.2 to 0.8 to balance excessive connectivity filtering for small values and noise persistence for large values. $\tau_{cr}$ is maintained within a confidence threshold range of 0.85 to 0.99 to avoid error correction issues. It can be observed that within a specific range, the influence of $\tau_{ns}$ and $\tau_{cr}$ on the final model's adaptability is minimal. Regarding $\tau_{ns}$: a larger $\tau_{ns}$ retains more connectivity but introduces more noise, leading to a decrease in adaptability. A smaller $\tau_{ns}$ maintains higher connectivity accuracy, but a lower quantity reduces richness and results in a decrease in mIoU. Regarding $\tau_{cr}$: a larger $\tau_{cr}$ corrects some confident connections, improving accuracy and adaptability. A smaller $\tau_{cr}$ introduces more noise, compromising accuracy and adaptability. The final $\tau_{ns}$ / $\tau_{cr}$ is set to 0.6 / 0.95.

**Why is it easier to filter noise at the connectivity level than at the pixel level?** We find that compared to pixel-level classification, connectivity-level classifiers can more easily construct a compact feature space, thus simplifying noise filtering. As shown in Fig. 7, we measure the distance from features to class cluster centers (introduced as the "FID" metric in [72]) for ADVENT, ProDA,

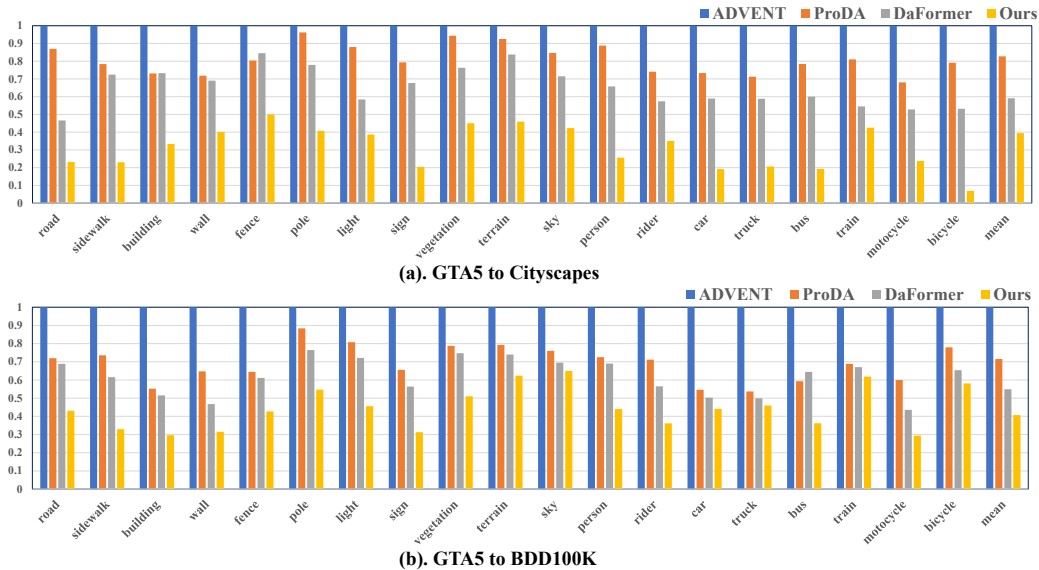

**(a). GTA5 to Cityscapes**

**(b). GTA5 to BDD100K**

Figure 7: Comparison of pixel-level class feature distribution and our connectivity-level feature distribution using the FID [72] metric. Data comes from (a) GTA5 → Cityscapes and (b) GTA5 → BDD100K experiment.

DAFormer, and our method after adaptation. We use the FID value for each category in the ADVENT method as the baseline, and normalize the FID values of other methods by dividing them by this baseline. Therefore, the FID values for all categories in ADVENT are set to 1. A smaller FID value indicates a more compact cluster and better feature separability. The results indicate that our connectivity-level classifier significantly enhances feature separability.

# E    More diverse scenes

We explore SeCo's performance in more segmentation scenarios, including indoor scenes, cross-domain medical images, and cross-domain remote sensing images, as shown in Table 9 - Table 11. Based on these positive experimental results, we believe SeCo has the potential to be integrated into more segmentation scenarios involving the use of unlabeled data.

*Indoor scenes*: The commonly used segmentation dataset for indoor scenes is ADE20K, but no cross-domain segmentation benchmark exists. Thus, we conduct experiments on a semi-supervised segmentation task, which also involves utilizing pseudo-labels from unlabeled data (domain adaptation is seen as semi-supervised learning with domain shift). We perform PSA using the stuff and thing definitions in ADE20K and execute SCC with default parameters. We use the Unimatch model as the baseline and follow its settings. The results of incorporating SeCo are shown in the table below. In multiple labeled data splits (1/64 - 1/8) in ADE20K, SeCo shows significant performance improvement compared to directly using SAM.

*Medical images*: We follow the medical image UDA setup from Sim-T[3], using the Endovis17 and Endovis18 abdominal surgery datasets collected from different devices containing 3 instrument type classes. We treat the segmentation objects as "things" and aggregate pixels using only boxes and points from the pseudo-label. The table below shows how SeCo greatly benefits SAM in this challenging task.

*Remote sensing*: We follow the UDA setup in remote sensing from the CIA[4], using the Potsdam and Vaihingen datasets collected from different satellites. These datasets contain five common semantic categories: car, tree, impervious surface, building, and low vegetation. We treat cars and buildings as "things," and the rest as "stuff." The table below shows that SeCo still achieves significant performance improvement compared to directly using SAM.

| Indoor Scenes: ADE20K | | | | |
|---|---|---|---|---|
| Methods | 1/64 | 1/32 | 1/16 | 1/8 |
| UniMatch [83] (CVPR'23) | 21.1 | 28.8 | 30.9 | 35.0 |
| Switch [53] (NIPS'23) | 22.6 | 27.9 | 30.1 | 33.8 |
| +SAM (SA) | 20.6 (-0.5) | 28.9 (+0.1) | 31.3 (+0.4) | 35.5 (+0.5) |
| +SeCo (w/o SCC) | 21.8 (+0.7) | 28.0 (+0.9) | 31.9 (+1.1) | 36.0 (+1.0) |
| +SeCo (Full) | **25.1 (+4.0)** | **32.4 (+3.6)** | **34.6 (+3.7)** | **38.1 (+3.1)** |

Table 9: The performance of indoor scenes on ADE20K.

| Medical Image: Endovis17→Endovis18 | | | | |
|---|---|---|---|---|
| Performance | scissor | needle driver | forceps | mIoU |
| SimT [18] (TPAMI'23) | 76.2 | 39.8 | 58.9 | 58.3 |
| +SAM | 73.0 (-3.2) | 38.3 (-1.6) | 55.7 (-3.2) | 55.6 (-2.7) |
| +SeCo (Full) | **78.4 (+2.2)** | **41.2 (+1.4)** | **61.2 (+2.3)** | **60.4 (+2.1)** |

Table 10: The performance of medical scene: Endovis17 → Endovis18 .

| Remote sensing: Potsdom → Vaihingen | | | | | | |
|---|---|---|---|---|---|---|
| Performance | Imp.Sur | Build. | Vege. | Tree | Car | mIoU |
| CIA-UDA [54] (TGARS'23) | 63.3 | 75.1 | 48.4 | 64.1 | 52.9 | 60.6 |
| +SAM (Semantic Alignment) | 61.78 (-1.6) | 70.67 (-4.4) | 50.1 (+1.7) | 66.84 (+2.7) | 50.9 (-2.0) | 60.1 (-0.5) |
| +SeCo (w/o SCC) | 64.37 (+1.0) | 76.31 (+1.2) | 50.45 (+2.1) | 66.41 (+2.3) | 54.67 (+1.8) | 62.4 (+1.8) |
| +SeCo (Full) | **69.47 (+6.1)** | **80.53 (+5.4)** | **51.97 (+2.5)** | **70.69 (+6.5)** | **57.73 (+4.6)** | **66.1 (+5.5)** |

Table 11: The performance of Remote sensing: Potsdom → Vaihingen.

## F  Visualization

Fig. 8 displays the pseudo-label outputs of PSA and SCC in the GTA5 → BDD-100K task. In this open compound adaptation task, the model's initial speckled pseudo-labels exhibit considerable noise. It is noticeable that PSA aggregates speckle noise into connected components, concurrently amplifying noisy pseudo-labels. Subsequently, SCC further suppresses and corrects the connected noise from PSA, leading to more structured and lower-noise pseudo-labels. This further validates the motivation behind the design of PSA and SCC.

Fig. 9 demonstrates that SeCo has the potential to filter out noisy semantic connectivity when faced with domain shifts in various domain environments. Compared to directly using SAM to complete pseudo-labels, SeCo provides a safer and more efficient approach, to some extent avoiding the problem of error accumulation in self-training.

Fig. 10 shows that for unseen domain environments simulated by Stable Diffusion, the pseudo-labels generated by basic domain generalization methods are very noisy and chaotic. In such cases, filtering at the pixel level is more challenging. As seen in (b), pixel-level filtering functions struggle to eliminate both closed-set and open-set noise. As seen in (c), directly using SAM to complete pseudo-labels easily leads to the propagation of noise. (d) shows that our SeCo has the potential to filter out some of the noisy labels in such wild data.

## G  Limitations

While our SeCo can integrate existing domain adaptation and domain generalization methods, it lacks an explicitly designed category-balanced connectivity noise filtering method. Additionally, the results demonstrate that SeCo can identify open-set noise in outdoor data. However, due to the current evaluation environment limitations, we cannot directly evaluate its effectiveness. In the future, SeCo is expected to incorporate more visual foundational models to enhance pseudo-labeling for outdoor data targeting unknown distributions.

## H  Broader Impacts

SeCo aims to enhance the adaptability of segmentation to unseen domains or target domains, and has the potential to be applied in open-world scenarios. It has the potential to be combined with

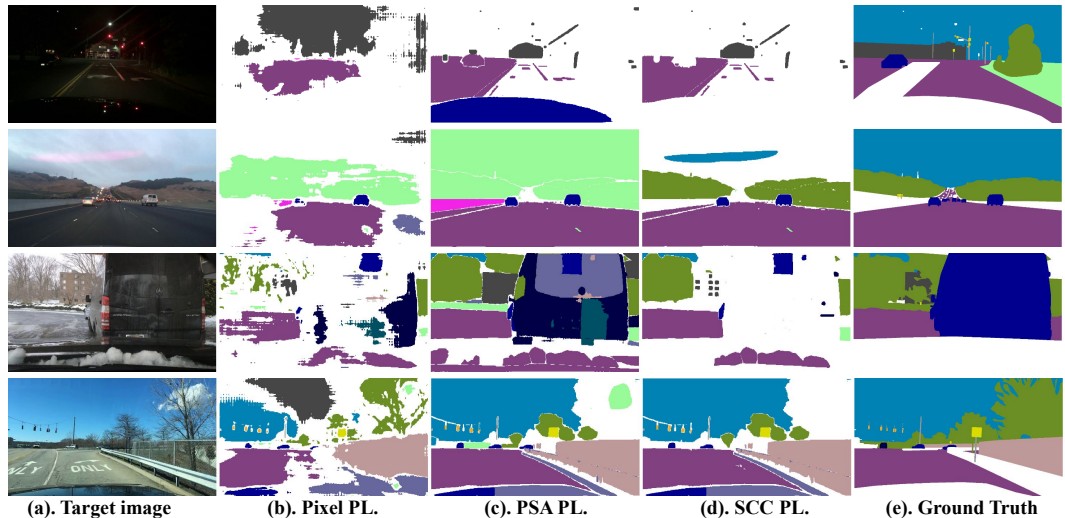

| (a). Target image | (b). Pixel PL. | (c). PSA PL. | (d). SCC PL. | (e). Ground Truth |

Figure 8: Comparison of pseudo-labels generated by original, PSA, and SCC in the Open Compound domain adaptation task GTA5 → BDD-100k. White regions in pseudo-label denote filtered areas.

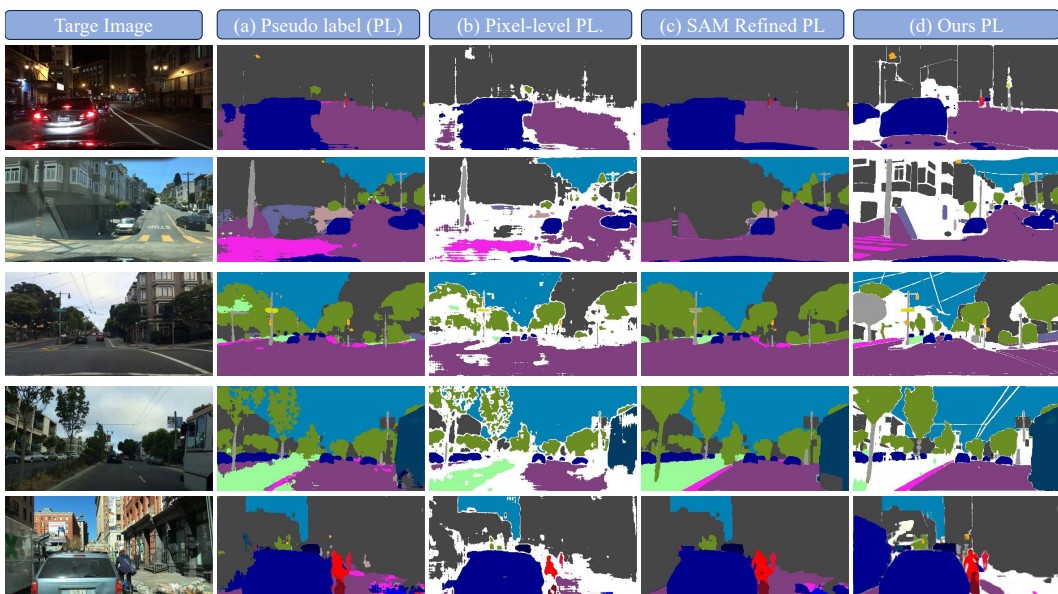

| Targe Image | (a) Pseudo label (PL) | (b) Pixel-level PL. | (c) SAM Refined PL | (d) Ours PL |

Figure 9: More visualization results of pseudo-labels from different methods on GTA5 → BDD-100k results.(a) Pseudo-Labels (PL), (b) pixel-level PL [39], (c) SAM-refined PL [4], and (d) the proposed connectivity-level PL. The white area in the PL represents the filtered area.

multi-modal large models to perform open-set adaptation in more complex open environments. In addition, SeCo has the potential to be applied in medical scenarios involving privacy or property rights protection.

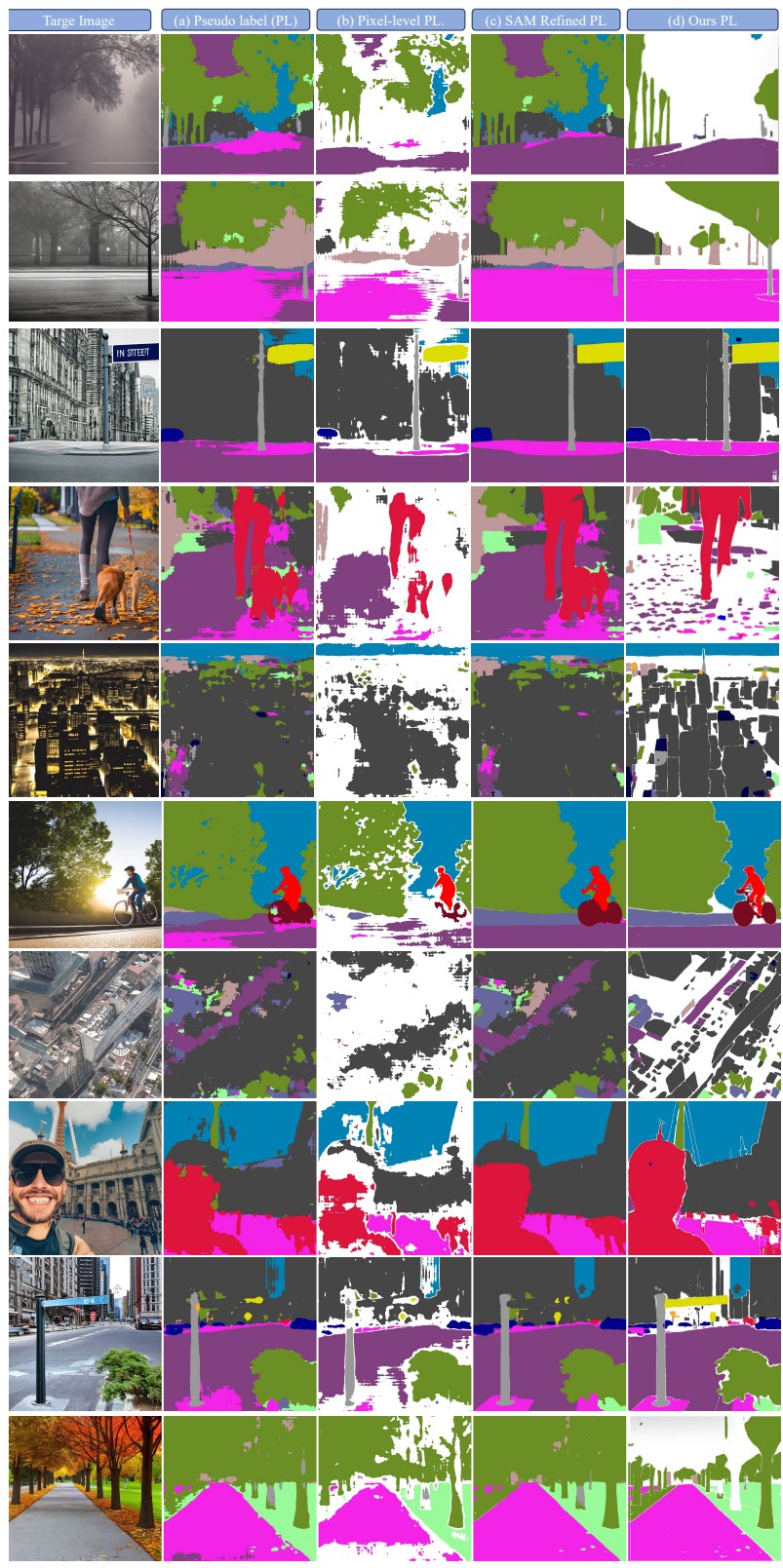

Figure 10: More pseudo-label visualizations on synthetic data from stable diffusion. It shows that SeCo has the potential to eliminate open-set noise. (a) Pseudo-Labels (PL), (b) pixel-level PL [39], (c) SAM-refined PL [4], and (d) the proposed connectivity-level PL. The white area in the PL represents the filtered area.

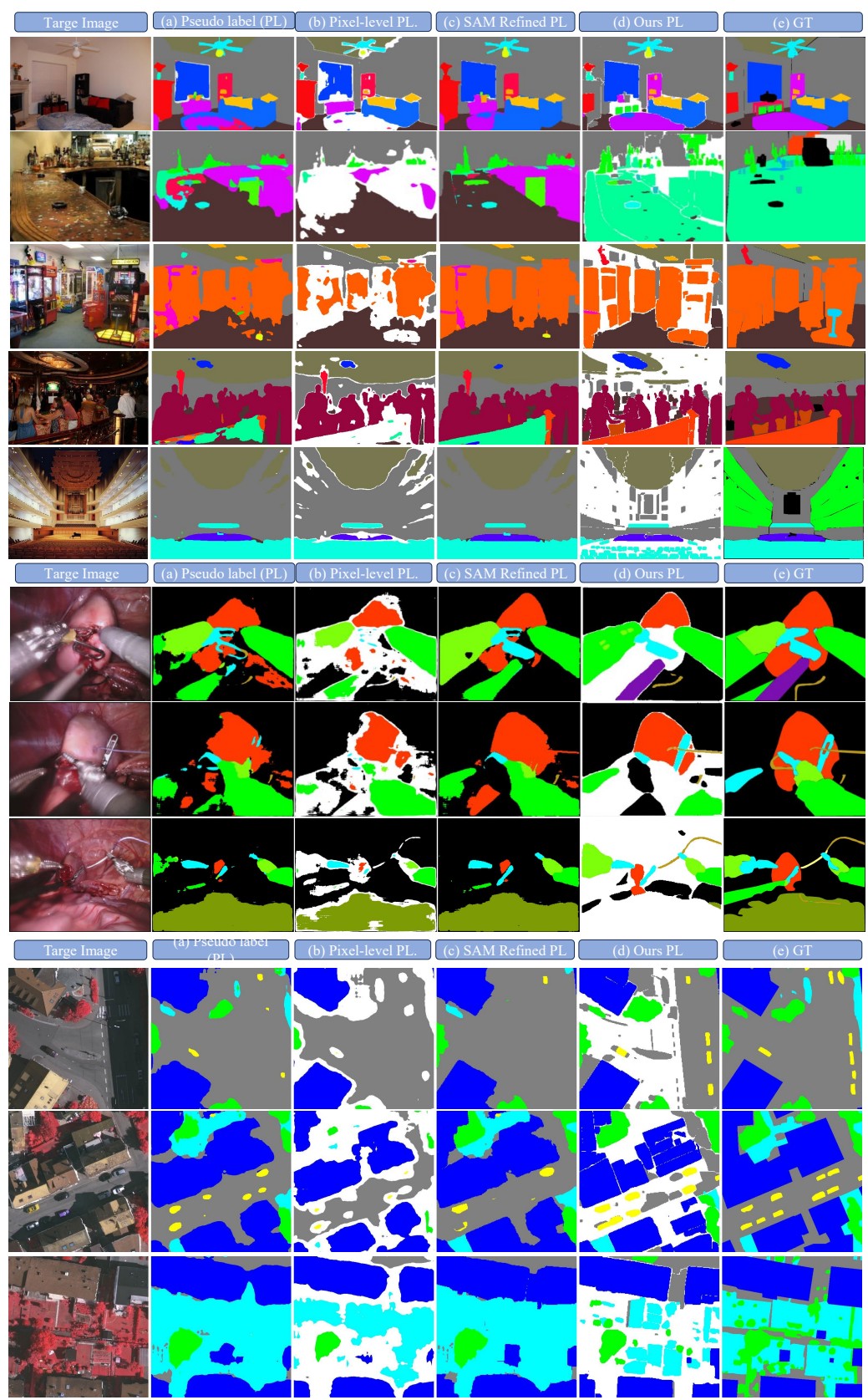

Figure 11: More pseudo-label visualizations on More diverse scenes. The white area in the PL represents the filtered area. Our method effectively filters out and corrects closed-set noise across multiple semantic segmentation scenes.

