# OpenReview forum: "Connectivity-Driven Pseudo-Labeling Makes Stronger Cross-Domain Segmenters"
_NeurIPS.cc/2024/Conference — NeurIPS 2024 poster_

### Official Review · Reviewer_pA2t · 2024-07-02

**Soundness:** 3
**Presentation:** 2
**Contribution:** 3
**Rating:** 5
**Confidence:** 4

**Summary:**

This paper proposes a method to improve the quality of pseudo labels for unlabeled target data. Specifically, it designs different strategies for using SAM to handle 'stuff' and 'things,' which are classified based on the initial pseudo labels. Subsequently, it fits a Gaussian Mixture Model (GMM) on the per-connectivity/component loss distribution to differentiate between noisy and clean components. The proposed method is evaluated on several cross-domain semantic segmentation tasks and demonstrates significant improvements.

**Strengths:**

1.  The design of pixel semantic aggregation is well-motivated and clearly ablated.

2.  The experiments are extensive, evaluating various cross-domain semantic segmentation tasks and different pseudo-labeling methods.

3.  The ablation studies clearly demonstrate the contributions of each module.

**Weaknesses:**

1. The novelty of the semantic connectivity correction (SCC) module is limited, as it trivially adapts the technique in DivideMix (ICLR 2020) to handle components instead of images.

2.  It would be beneficial to discuss related work on noisy label learning since the second part (SCC) focuses on noisy label recognition and correction, which are common in noisy label learning.

3. The use of domain generalization (DG) with synthesized data is not a common setting and may not accurately represent DG. Additionally, Eq. 1 is not suitable if including DG settings as x_t is not accessible.

4. The implementation details of CLIP+SAM in Line 248 are unclear.

5.  In Table 6, it would be helpful to ablate the recognition and correction steps in SCC. What would the performance be without correcting the noise (Eq. 4)?

**Questions:**

Please refer to the weaknesses section.

**Limitations:**

The authors have discussed the limitations and potential impact of the proposed method.

---

> ### Author Rebuttal · Authors · 2024-08-03
>
> ### Thank you for recognizing our work and providing constructive feedback.
>
> ### Q1: SCC vs DivideMix
>
> Although our SCC is partly inspired by DivideMix, it is different from DivideMix from four aspects.
>
> #### Address Different Tasks:
> - **SCC**: Cross-domain semantic segmentation.
> - **DivideMix**: Noisy labels learning for image classification.
>
> #### Solve Different Problems:
> - **SCC**: We focus on mitigating the **pixel-level noises** in pseudo-labels raised by domain shifts and SAM refinement.
> - **DivideMix**: It focuses on mitigating the **image-level label noises**.
>
> #### Different Motivations:
> - **SCC**: We aim to convert pixel-level denoising to connectivity-level denoising, which takes the advantage of context information (relationship among neighbouring pixels) and thus is more robust to noises.
> - **DivideMix**: It aims to effectively address the image-level denoising.
>
> #### Different Implementations:
> - **SCC**:
>   0. Use pooling to treat aggregated pixels as **connectivity**.
>   1. Use **early stop and GMM** for filtering and correcting connectivity with noise.
>   2. Use a **fixed threshold** to assign pseudo-label for selected connectivities.
>   3. Use **connectivities with pseudo-labels** to train the model.
> - **DivideMix**:
>   1. Use **co-divide-based GMM** to divide data.
>   2. Use **co-guess** for pseudo-label correction.
>   3. Use **mixmatch** for semi-supervised learning.
>
> To better verify the effectiveness of our SCC, we add two experiments: a) directly applying DivideMix to pixel-level denoising. b) using DivideMix to denoise the pixels aggregated by our PSA (using SAM).  The results show that pixel-level denoising methods based on DivideMix are ​​inferior to SCC even with SAM, highlighting the advantage of denoising at the connectivity level.
>
> To this end, we hope Reviewer pA2t can find that our SCC is different from DivideMix and that the proposed connectivity-level denoising is a key step in our full framework.
>
> |  | GTA → Cityscapes (UDA) | SYNTHIA → Cityscapes (UDA) | GTA5 → BDD-100k (OC-DA)  |
> |:-:|:-:|:-:|:-:|
> | ProDA (CVPR'21) | 53.7| 51.9| 41.7|
> | DivideMix | 49.8 | 47.6  | 37.4   |
> | PSA+DivideMix | 60.1   | 53.4 | 44.2  |
> | SeCo  | **64.1** | **58.6**| **49.5** |
>
> ### Q2: Related work on noisy label learning.
>
> Good suggestion. We add a discussion on Noisy Label Learning(NLL).  Currently, NLL focuses on classification tasks with techniques like robust loss design [1], regularization [2], label weighting [3], and correction [4]. These methods typically target image-level noise and may not be effective for pixel-level segmentation, which involves complex spatial and semantic dependencies among pixels. Maintaining spatial consistency across millions of pixels is a challenge for current image-level denoising methods.  In segmentation, few methods focus on denoising the pseudo-label, such as ProDA [84] and RPL [90], which denoise each pixel independently and still face the challenges highlighted in our paper.  Our SeCo effectively links image-level techniques with segmentation tasks, offering novel solutions for pseudo-label denoising in segmentation. In the future version, we will add more related work to this section.
>
> ### Q3: DG experiment.
>
> In this experiment, we followed CLOUDS (CVPR'24), which uses synthetic data from Stable Diffusion(SD) to assist DG. Our aims are: 1) to show that our method can handle challenging synthesized data with open-set noise, and 2) to compare our SAM usage with the competitive scheme in CLOUDS. We agree with your point that using synthetic data from SD does not constitute a strict DG setup. In the revised version, we will clarify that this experiment is to assist DG rather than adhering to a strict DG setting.
>
> ### Q4: The details of CLIP+SAM in Line 248.
>
> For CLIP+SAM, the steps are: 1) Use SAM to segment the input image. 2) Extract the largest bounding rectangle from each segment.
> 3) Create text descriptions for categories, e.g., "a photo of a road." 4) Use CLIP to match image patches with text descriptions, assigning text labels as categories.
> CLIP+SAM+UDA combines pseudo-labels from CLIP+SAM and UDA, using voting fusion to select consistent predictions for training. These details will be added to the main text.
>
> ### Q5: More detailed ablation on SCC.
>
> We conduct a detailed ablation study on SCC across multiple tasks in GTA $\rightarrow$ Cityscapes, evaluating it with two metrics: PL mIoU (pseudo-label quality on the training set) and Val. mIoU (model performance on the validation set). The results show that: 1). removing the filter function causes an average loss of 4.1% mIoU in pseudo-label quality and 1.8% mIoU in validation performance. 2). removing the correction function causes an average loss of 3.6% mIoU in pseudo-label quality and 1.2% mIoU in validation performance. This shows that both modules are crucial for reducing pseudo-label noise and improving model performance.
>
> | Prompt Way          | w/o SAM |           | PSA     |           | SCC w/o correcting |           | SCC w/o filter |           | SCC     |            |
> |-|-|-|-|-|-|-|-|-|-|-|
> |                     | PL mIoU | Val. mIoU | PL mIoU | Val. mIoU | PL mIoU            | Val. mIoU | PL mIoU        | Val. mIoU | PL mIoU | Val. mIoU  |
> | SeCo+ProDA (UDA)    | 69.3  | 53.7 | 76.8    | 62.1 | 80.1 | 63.1| 79.1 | 62.9 | **81.9**| **64.1**|
> | SeCo+DAFormer (UDA) | 76.9 | 68.2 | 81.9    | 70.3| 85.9  | 72.1 | 83.7 | 71.9| **88.6** | **73.4** |
> | SeCo+DTST (SF-UDA)  | 68.1| 52.1 | 76.2    | 57.9| 77.8 | 59.1| 77.9| 58.7  | **80.1**  | **60.5**  |
> | SeCo+BiMem(BB-UDA)  | 57.9 | 48.2 | 67.5    | 54.4  | 70.5  | 55.6  | 68.8 | 55.7| **72.6**    | **56.7** |
>
>
> [1]. Generalized cross entropy loss for training deep neural networks with noisy labels, (NIPS'18)
>
> [2]. Robust early-learning: Hindering the memorization of noisy labels, (ICLR'21)
>
> [3]. Meta pseudo labels, (CVPR'21)
>
> [4]. Dividemix: Learning with noisy labels as semi-supervised learning (ICLR'21)

---

> > ### Comment · Reviewer_pA2t · 2024-08-12
> > **Thank you for your response**
> >
> > Thank you to the authors for providing the rebuttal, including the additional ablation study and explanations. Most of my concerns have been addressed. Regarding the novelty of SCC, I agree with the authors that the tasks are different. However, the fundamental challenge of noisy label correction remains the same, and the technical novelty of SCC appears to be quite subtle. Additionally, when comparing with DivideMix, I believe it would be more convincing to apply DivideMix at the connectivity level. I would like to maintain my initial rating.

---

> ### Author Response · Authors · 2024-08-12
> **Reply to Reviewer pA2t - A more detailed comparison of SCC and Dividemix**
>
> Thank you for your feedback, it is very helpful for us to improve our paper.
>
> First, we would like to emphasize that one of the main contributions of SCC is to provide the idea of ​​denoising at the connectivity level, which makes it possible to apply other image-level denoising methods such as Dividemix to segmentation tasks.
> Building on this, we further provide the implementation of DivideMix at the connectivity level across multiple tasks, including GTA → Cityscapes (G→C UDA), SYNTHIA → Cityscapes (S→C UDA), GTA5 → BDD-100k (G→B OC-DA), Endovis17→Endovis18 (E17→E18) and Potsdom → Vaihingen (P→V).
>
> **Implementation comparison**: In DivideMix, two models are first trained with different warm-up iterations. Then, each model uses different GMMs to select samples and then exchanges the selected samples (co-divide) between the two models for further co-guessing-based training with more iterations. The implementation requires careful selection of several hyperparameters, including the warm-up iterations for both models, the number of further training iterations, and the threshold for co-guessing. In contrast, SCC requires only a single model with fixed 5k early-stopping training for all tasks and then uses GMM for sample selection.
>
> **Performance comparison**:  The performance and training time of DivideMix at the connectivity level are shown in the table below. We observe that DivideMix performs well in tasks with lower noise rates, such as GTA → Cityscapes (UDA), where it achieves comparable performance to SCC.  However, in tasks with higher initial noise rates, such as GTA5 → BDD-100k (OC-DA), SCC outperforms DivideMix. This is because higher noise rates introduce more challenges in hyperparameter tuning, particularly in choosing the warm-up epochs for the two models, e.g., close warm-up iterations fail to create sufficiently different models, while significantly different iterations can lead to one model overfitting to noise. (The table below shows results where we chose models with 2.5K and 5K iterations for both models.) This complexity hinders DivideMix from maintaining stable denoising performance in more challenging scenarios.
>
> **Overall, compared to DivideMix, we think our SCC not only introduces a new idea of denoising at the connectivity level,  but offers a simpler implementation, requires less hyperparameter tuning, has shorter denoising training times, and delivers more stable denoising performance**.
>
> | | Urban Scene            | | || |  | Medical              | | Remote Sensing      | |
> |:--:|:--:|:--:|:--:|:--:|:--:|:--:|:--:|:--:|:--:|:--:|
> |                            | G→ C |                    | S→ C|                    | G→ B |                    |  E17→E18 |                    | P→ V| |
> | Methods                    | Val mIoU               | Training Time (/h) | Val mIoU                   | Training Time (/h) | Val mIoU                | Training Time (/h) | Val mIoU             | Training Time (/h) | Val mIoU            | Training Time (/h)  |
> | PSA+**DivideMix-Pixel**        | 60.1                   | 10                 | 53.4                       | 10                 | 44.2                    | 12                 | 47.9                 | 8                  | 50.1                | 8                   |
> | PSA+**DivideMix-Connectivity** | 63.9                   | 6                  | 57.3                       | 6                  | 47.7                    | 7                  | 58.7                 | 6                  | 65.3                | 6                   |
> | PSA+**SCC**                       | **64.1 (+0.2)**            | 2.5                  | **58.6 (+1.3)**                | 2.5                  | **49.5 (+1.8)**             | 2.5                  | **60.4 (+1.7)**                 | 1.5                | **66.1 (+0.8)**               | 1.5                 |

---

### Official Review · Reviewer_hfgi · 2024-07-11

**Soundness:** 3
**Presentation:** 3
**Contribution:** 3
**Rating:** 6
**Confidence:** 3

**Summary:**

This paper proposed an effective method to generate reliable high-quality pseudo-labels for cross-domain semantic segmentation. It introduces a novel method, Semantic Connectivity-driven Pseudo-labeling (SeCo), which addresses these issues by formulating pseudo-labels at the connectivity level, thereby improving noise localization and correction. The SeCo method consists of two components: Pixel Semantic Aggregation (PSA), which aggregates speckled pseudo-labels into semantic connectivity, and Semantic Connectivity Correction (SCC), which corrects connectivity noise through a classification task guided by loss distribution. Extensive experiments show that SeCo significantly enhances the performance of existing methods across various semantic segmentation tasks.

**Strengths:**

### Clear Motivation and Well-Proposed Methods
- The limitations of existing pseudo-labeling methods (Figure 1) are well-described and make sense.
- Utilizing the SAM to refine pseudo-labels by splitting into stuff and things categories is convincing.
- The early learning in noisy label learning in SCC is interesting and effective.

### Comprehensive Experiments and Analysis
- This paper provided extensive experimental results to verify the effectiveness of the proposed method, achieving remarkable improvements.
- The proposed method is well-ablated including a comparison with the SAM-guided method (COLUDS [4]) and various strategies for employing SAM (Table 5).

**Weaknesses:**

### More detailed quantitative results

- It would be great if the authors could provide quantitative analysis when using prompting only or semantic alignment only without separating things and stuff.
- It would be helpful if the authors could provide the effect of the enlargement factor for the bounding box area, which is set to 1.5 by default.
- It would be better if the authors could provide the effect of the number of training iterations for the connectivity classifier.

**Questions:**

Overall, I like the concept of the proposed method and its effectiveness.
My initial recommendation is Weak Accept.
However, since I am unfamiliar with this research field, I will finalize the rating after discussion with other reviewers.

**Limitations:**

The authors have discussed the limitations.

---

> ### Author Rebuttal · Authors · 2024-08-03
>
> ### Thank you for your recognition of our work and your constructive suggestions.
>
> ### Q1: Quantitative Analysis on "Prompting Only" and "Semantic Alignment"
>
> We conduct ablation studies on "Prompting Only" (PO) and "Semantic Alignment"(SA) across multiple tasks in GTA $\rightarrow$ Cityscape. We provide two metrics for these detailed ablations: PL mIoU (pseudo-label quality on the training set) and Val. mIoU (model performance on the validation set after training with those pseudo-labels).  As shown in the table below, the "prompting only" method reduces the quality of pseudo-labels in the training set, leading to poor adaptation performance. This is because the unreliable interaction method introduces excessive noise into the pseudo-labels generated by SAM. "Semantic alignment" improves the quality of the training set pseudo-labels, but the improvement is limited, resulting in limited adaptation benefits. In contrast, our method enhances the quality of the training set pseudo-labels through better interaction, leading to superior performance gains.
>
> | Prompt Way           | w/o SAM |   w/o SAM     | PO |    PO   | SA  |   SA    | PSA  (Ours)      |   PSA  (Ours)    |
> |:--------------------:|:--------------:|:-----:|:--------------:|:-----:|:------------------:|:-----:|:---------:|:-----:|
> |                      | PL mIoU        | Val. mIoU | PL mIoU        | Val. mIoU | PL mIoU            | Val. mIoU | PL mIoU   | Val. mIoU |
> | SeCo+ProDA (UDA)     | 69.3           | 53.7  | 61.8 (-7.5)    | 48.0 (-5.7) | 73.4 (+4.1)        | 60.9 (+7.2) | **76.8 (+7.5)** | **62.1 (+8.4)** |
> | SeCo+DAFormer (UDA)  | 76.9           | 68.2  | 70.1 (-6.8)    | 64.6 (-3.6) | 79.7 (+2.8)        | 69.7 (+1.5) | **81.9 (+5.0)**|**70.3 (+2.1)** |
> | SeCo+DTST (SF-UDA)   | 68.1           | 52.1  | 62.5 (-6.0)    | 46.7 (-5.4) | 72.5 (+4.4)        | 56.1 (+4.0) | **76.2 (+8.1)** | **57.9 (+5.8)** |
> | SeCo+BiMem (BB-UDA)  | 57.9           | 48.2  | 51.8 (-6.1)    | 42.8 (-5.4) | 62.1 (+4.2)        | 54.4 (+6.2) | **67.5 (+9.6)** | **54.4 (+6.2)** |
>
> ### Q2: The Effect of the Enlargement Factor on the Bounding Box Area
>
> As shown in the table below, we validate the impact of bounding box size on adaptation results across multiple UDA settings. Overall, a too-small bounding box (e.g., 0.5) causes a performance drop because it fails to encompass the "things" category adequately, hindering the effectiveness of SAM prompts. Too-large bounding boxes (e.g., 5) result in a slight performance decline due to poor delineation of the "things" category, leading to semantic ambiguity in SAM. When the bounding box size is between 1.5 to 2 times the size of the largest external rectangle, the model achieves the best adaptation performance, with minimal performance variation within this range.
>
> | Bounding Box Area    | 0.5   | 1     | 1.5   | 2     | 5     |
> |:---------------------:|:-----:|:-----:|:-----:|:-----:|:-----:|
> | GTA → Cityscapes (UDA)     | 70.2  | 72.4  | **73.4**  | 73.1  | 72.1  |
> | SYNTHIA → Cityscapes (UDA) | 61.5  | 64.2  | **65.1**  | 65.0  | 63.7  |
> | GTA5 → BDD-100k (OC-DA)    | 47.1  | 48.5  | 49.5  | **49.9**  | 48.3  |
>
> ### Q3: The Effect of the Number of Training Iterations on the Connectivity Classifier.
>
> We supplement our study with the impact of early-stop iterations on adaptation performance across multiple tasks in GTA $\rightarrow$ Cityscapes. As shown in the table below, both too few iterations (1K) and too many iterations (20K) harm adaptation performance. Too few iterations result in the classifier's insufficient fit to the connectivity, while too many iterations cause overfitting to noise in the connectivity. Setting the iterations between 5K and 10K provides stable denoising benefits and performance improvements.
>
> | Training Iterations | 1K   | 2K   | 5K   | 10K  | 20K  |
> |:-------------------:|:----:|:----:|:----:|:----:|:----:|
> | SeCo+ProDA (UDA)    | 62.1 | 63.7 | 64.1 | **64.4** | 63.6 |
> | SeCo+DAFormer (UDA) | 71.9 | 73.0 | 73.4 | **73.7** | 72.7 |
> | SeCo+DTST (SF-UDA)  | 57.5 | 60.3 | **60.5** | 60.1 | 59.1 |
> | SeCo+BiMem (BB-UDA) | 54.6 | 56.3 | **56.7** | 55.9 | 55.7 |

---

### Official Review · Reviewer_SXcd · 2024-07-13

**Soundness:** 3
**Presentation:** 3
**Contribution:** 3
**Rating:** 5
**Confidence:** 4

**Summary:**

This paper tackles the cross-domain semantic segmentation problem. It incorporates two modules, including the Pixel Semantic Aggregation and the Semantic Connectivity Correctio modules. The former adopts the SAM model to separately refine pseudo-labels for thing and stuff classes. The latter adopts the refined masks to train a connectivity classifier to further rectify pseduo-labels. Detailed experiments are provided under both the domain adaptation and the domain generalization settings. Results demonstrate that the method can effectively enhance the pseudo-label's quality and is compatible with various baseline methods.

**Strengths:**

- The paper is well-written and easy to follow. The figures are clear and helpful to the reader's understanding.
- The method is well-motivated and sound. The technical details are provided.
- There are plenty of analytic experiments. The results are impressive in that the proposed method can effectively enhance the performance with various baseline methods. The experiments in Table 5 conducted a detailed analysis concerning the SAM.

**Weaknesses:**

- The work may be limited to a relatively narrow field. Conducting experiments with the GTA5/Cityscape/SYNTHIA datasets has been the standard for cross-domain segmentation issues for a long while. Whether the proposed method can take effect in broader segmentation tasks concerning indoor scenes, medical images, and more categories is unsure.
- Though effective, the SAM model is pretrained with a huge amount of data associated with precise mask labels. Therefore, comparing previous methods without this well pretrained segmentation model risks fairness concerns.

**Questions:**

Please see the weaknesses for details. Overall, this paper does not have obvious drawbacks to me, though its impact may be limited. Therefore, the reviewer recommends a borderline accept.

**Limitations:**

The authors adequately discussed limitations in the appendix.

---

> ### Author Rebuttal · Authors · 2024-08-03
>
> ###  Thank you for your recognition and your useful suggestions.
> ### Q1. The work may be limited to a relatively narrow field.
>
> Good suggestion. We explore SeCo's performance in more segmentation scenarios, including indoor scenes, cross-domain medical images, and cross-domain remote sensing images, as shown in the Table below. Based on these positive experimental results, we believe SeCo has the potential to be integrated into more segmentation scenarios involving the use of unlabeled data.
> **For a qualitative visual comparison, see the PDF in Rebuttal**.
>
> 1. Indoor scenes: The commonly used segmentation dataset for indoor scenes is ADE20K, but no cross-domain segmentation benchmark exists. Thus, we conduct experiments on a semi-supervised segmentation task, which also involves utilizing pseudo-labels from unlabeled data (domain adaptation is seen as semi-supervised learning with domain shift). We perform PSA using the stuff and thing definitions in ADE20K and execute SCC with default parameters. We use the Unimatch model as the baseline and follow its settings. The results of incorporating SeCo are shown in the table below. In multiple labeled data splits (1/64 - 1/8) in ADE20K, SeCo shows significant performance improvement compared to directly using SAM.
> | Indoor Scenes: ADE20K |             |             |             |              |
> |:-:|:-:|:-:|:-:|:-:|
> | Labeled data  | 1/64        | 1/32   | 1/16    | 1/8    |
> | UniMatch [1] (CVPR'23) | 21.1   | 28.8   | 30.9  | 35.0    |
> | Switch [2] (NIPS'23)  | 22.6   | 27.9    | 30.1   | 33.8  |
> | +SAM (Semantic Alignment)   | 20.6 (-0.5) | 28.9 (+0.1) | 31.3 (+0.4) | 35.5 (+0.5)  |
> | +SeCo (w/o SCC)    21.8 (+0.7) | 28.0 (+0.9) | 31.9 (+1.1) | 36.0 (+1.0)  |
> | +SeCo (Full)  | **25.1 (+4.0)** | **32.4 (+3.6)** | **34.6 (+3.7)** | **38.1 (+3.1)**  |
>
> 2. Medical images: We follow the medical image UDA setup from Sim-T[3], using the Endovis17 and Endovis18 abdominal surgery datasets collected from different devices containing 3 instrument type classes. We treat the segmentation objects as "things" and aggregate pixels using only boxes and points from the pseudo-label. The table below shows how SeCo greatly benefits SAM in this challenging task.
> | Medical Image:  Endovis17→Endovis18 |             |               |             |              |   |
> |-|- |- |-|-|---|
> | Performance | scissor     | needle driver | forceps     | mIoU         |   |
> | SimT [3] (TPAMI'23)  | 76.2        | 39.8          | 58.9        | 58.3         |   |
> | +SAM  (Semantic Alignment) | 73.0 (-3.2) | 38.3 (-1.6)   | 55.7 (-3.2) | 55.6 (-2.7)  |   |
> | +SeCo (Full)   | **78.4 (+2.2)** | **41.2 (+1.4)** | **61.2 (+2.3)** | **60.4 (+2.1)**  |   |
>
> 3. Remote sensing: We follow the UDA setup in remote sensing from the CIA[4], using the Potsdam and Vaihingen datasets collected from different satellites. These datasets contain five common semantic categories: car, tree, impervious surface, building, and low vegetation. We treat cars and buildings as "things," and the rest as "stuff." The table below shows that SeCo still achieves significant performance improvement compared to directly using SAM.
> | Remote sensing: Potsdom → Vaihingen |              |              |               |              |              |              |
> |:-----------------------------------:|:------------:|:------------:|:-------------:|:------------:|:------------:|:------------:|
> | Performance                         | Imp.Sur      | Build.       | Vege.         | Tree         | Car          | mIoU         |
> | CIA-UDA [4] (TGARS'23)              | 63.3         | 75.1         | 48.4          | 64.1         | 52.9         | 60.6         |
> | +SAM  (Semantic Alignment)          | 61.8 (-1.6)        | 70.67 (-4.4) | 50.1 (+1.7)   | 66.84 (+2.7) | 50.9 (-2.0)  | 60.1 (-0.5)  |
> | +SeCo (w/o SCC)                     | 64.4  (+1.0)       | 76.31 (+1.2) | 50.45 (+2.1)  | 66.41 (+2.3) | 54.67 (+1.8) | 62.4 (+1.8)  |
> | +SeCo (Full)                        | **69.4 (+6.1)** | **80.5 (+5.4)** | **51.9 (+2.5)**  | **70.6 (+6.5)** | **57.7 (+4.6)** | **66.1 (+5.5)**  |
>
>
> ### Q2: Fairness concerns
>
> We discuss fairness comparisons in detail in lines 185-195 of the original paper. We want to emphasize the following two points here:
> 1. To ensure a fair comparison, we conducted experiments without SAM, as shown in Figure 5 of Section 3.2. In these experiments, we applied our SeCo to the outputs from the source model's connectivity, not using SAM. Across multiple tasks and frameworks, this still shows a clear advantage over pixel-level self-training. This experiment demonstrates that the idea of connectivity denoising is universal. Even with less structured connectivity, it can alleviate the noise issue in self-training for semantic segmentation tasks. Moreover, well-structured connectivity can further enhance the performance of connectivity denoising.
> 2. Our method aims to be integrated to enhance existing pseudo-labeling methods rather than to compete with them. Results show that our method can significantly improve previous state-of-the-art baselines by adding our method to each baseline individually, which could be regarded as a fair comparison and shows the complementary of our method to these baselines.
>
> [1]. Revisiting weak-to-strong consistency in semi-supervised semantic segmentation CVPR'23
>
> [2]. Switching temporary teachers for semi-supervised semantic segmentation NIPS'23
>
> [3]. SimT: Handling Open-set Noise for Domain Adaptive Semantic Segmentation TPAMI'23
>
> [4]. Category-Level Assignment for Cross-Domain Semantic Segmentation in Remote Sensing Images. TGARS'23

---

> > ### Comment · Reviewer_SXcd · 2024-08-13
> >
> > Thanks for the response and efforts to provide additional information. It addressed most of my concerns. I'll maintain my positive score.

---

### Author Rebuttal · Authors · 2024-08-07

We sincerely thank the AC and the reviewers for their tremendous effort in handling our paper.

We have adequately addressed all the issues raised by the reviewers. These include providing validation in more segmentation scenarios (Reviewer #SXcd), explaining fairness concerns(Reviewer #SXcd), discussing hyperparameters, and conducting more ablation studies (Reviewers #hfgi, #pA2t), comparing with related work on learning with noisy labels (Reviewer #pA2t), and providing more detailed experimental descriptions (Reviewer #pA2t).

Recognized strengths of the paper by the reviewers:

- Clear motivation and convincing (Reviewers #SXcd, #hfgi, #pA2t)
- The paper proposes a novel connectivity-based denoising idea, which is interesting and effective (Reviewer #hfgi)
- The paper is well-written and easy to understand (Reviewers #SXcd, #hfgi, #pA2t)
- Comprehensive experiments and analysis, as well as detailed ablation experiments (Reviewers #SXcd, #hfgi, #pA2t)

We hope that the AC and the reviewers will consider the following factors when making the final decision:
1. The novel connectivity-based denoising framework for cross-domain segmentation, with reviewers confirming the rationality of the idea and its significant effectiveness in multiple tasks;
2. The comprehensive responses to all reviewer comments;
3. The open-source code provided to the reviewers (in the supplementary materials).

If you have any further questions or concerns, please let us know. We are happy to provide clarifications.

Authors of submission #5556

---

### Decision · Program_Chairs · 2024-09-25

**Decision:**

Accept (poster)

**Comment:**

The reviewers unanimously appreciated that the proposed method is well motivated and technically sound, and that the experiments are extensive. However, they at the same time raised concerns with potentially limited application scenarios (SXcd), unfair comparisons with previous work due to the use of SAM (SXcd), incremental novelty of the semantic connectivity correction module (pA2t), unclear part of the manuscript (pA2t), and lack of discussions on closely related work (pA2t); they also suggested additional in-depth analysis (hfgi). The authors' rebuttal and subsequent responses in the discussion period address these concerns and suggestions, and consequently, all the reviewers were supportive of the paper after the discussion period.

The AC agrees with the reviewers and recommends acceptance. The authors are strongly encouraged to carefully revise the paper to reflect the valuable comments by the reviewers, to add new results and discussions brought up in the rebuttal and discussions, and to further improve the quality of writing.